# A systematic review of the burden of, access to services for and perceptions of patients with overweight and obesity, in humanitarian crisis settings

Thomas Shortland[1], Majel McGranahan[1], Daniel Stewart[2], Oyinlola Oyebode[1], Saran Shantikumar[1], William Proto[1], Bassit Malik[1], Roger Yau[1], Maddie Cobbin[1], Ammar Sabouni[3], Gavin Rudge[4], Farah Kidy[1]*

1 Warwick Medical School, University of Warwick, Coventry, United Kingdom, 2 National Public Health Specialty Training Programme, South West Training Scheme, Bristol, United Kingdom, 3 Syria Development Centre, London, United Kingdom, 4 Institute of Applied Health Research, College of Medical and Dental Sciences, University of Birmingham, Birmingham, United Kingdom

* f.kidy@warwick.ac.uk

**Data Availability Statement:** All relevant data are within the manuscript and its Supporting Information files.

## Abstract

### Introduction

Excess body weight causes 4 million deaths annually across the world. The number of people affected by humanitarian crises stands at a record high level with 1 in 95 people being forcibly displaced. These epidemics overlap. Addressing obesity is a post-acute phase activity in non-communicable disease management in humanitarian settings. Information is needed to inform guidelines and timing of interventions. The objective of this review was to explore the prevalence of overweight and obesity in populations directly affected by humanitarian crises; the cascade of care in these populations and perceptions of patients with overweight and obesity.

### Methods

Literature searches were carried out in five databases. Grey literature was identified. The population of interest was non-pregnant, civilian adults who had experience of humanitarian crises (armed conflict, complex emergencies and natural disasters). All study types published from January 1st, 2011, were included. Screening, data extraction and quality appraisal were carried out in duplicate. A narrative synthesis is presented.

### Results

Fifty-six reports from forty-five studies were included. Prevalence estimates varied widely across the studies and by subgroups. Estimates of overweight and obesity combined ranged from 6.4% to 82.8%. Studies were heterogenous. Global distribution was skewed. Increasing adiposity was seen over time, in older adults and in women. Only six studies were at low risk of bias. Body mass index was the predominant measure used. There were no studies reporting cascade of care. No qualitative studies were identified.

**Funding:** FK is supported by an NIHR Doctoral Research Fellowship (grant number 300688) and SS is supported by an NIHR Clinical Lectureship. The views expressed are those of the authors and not necessarily those of the NIHR or the Department of Health and Social Care. The funders had no role in the study design, data collection and analysis, decision to publish, or preparation of the manuscript. https://www.nihr.ac.uk/.

**Competing interests:** The authors have declared that no competing interests exist.

## Conclusion

Overweight and obesity varied in crisis affected populations but were rarely absent. Improved reporting of existing data could provide more accurate estimates. Worsening obesity may be prevented by acting earlier in long-term crises and targeting risk groups. The use of waist circumference would provide useful additional information. Gaps remain in understanding the existing cascade of care. Cultural norms around diet and ideal body size vary.

## Introduction

Costing 2.8% of the world's gross domestic product, affecting over 2 billion people worldwide and causing 4 million deaths annually, excess body weight (including overweight and obesity) is a global health emergency [1]. The World Health Organisation (WHO) identifies excess body weight as a key risk factor for noncommunicable diseases (NCDs) [2]. Globally, more than 15 million die prematurely due to NCDs [2]. Reducing obesity can decrease premature mortality [3], thereby directly contributing to Sustainable Development Goal (SDG) 3.4 [4].

The potential negative impacts of obesity and overweight are not restricted to NCDs. As the global community has learnt over the course of the Covid-19 pandemic, increased adiposity is also a risk factor for morbidity and mortality caused by some infectious diseases [5].

The number of people affected by humanitarian crises including violence, persecution, natural disasters and human rights violations has increased steadily since 2010 and now stands at record high level with 82.4 million people being forcibly displaced at the end of 2020 [6]. The majority of those displaced have remained in their own countries (internally displaced people [IDPs]) and following Colombia, the most affected countries are in Africa and the Middle East. As for those displaced across borders, approximately two thirds come from Syria, Venezuela, Afghanistan, South Sudan and Myanmar [6].

In many of these countries NCDs are now more significant causes of death and disability than communicable diseases and levels of obesity and overweight are increasing [7–9].

Whilst in the past the issue of NCDs in humanitarian crises was largely forgotten, the increasingly overlapping nature of these epidemics is now recognised. There have been calls from practitioners and patients to increase research [10] and to improve prioritisation, recognition, prevention and management of NCDs in these settings [11–14]. An informal working group chaired by the United Nations High Commissioner for Refugees (UNHCR) and with membership of academics, policy makers, WHO and key non-governmental organisations (NGOs) is leading the way on delineating operational considerations for NCD management in humanitarian settings [15]. Obesity features in these discussions as a risk factor for chronic diseases to be addressed after the acute phase of the crisis.

To inform these developments, there are a suite of systematic reviews which bring together the evidence on diabetes [16, 17], substance misuse [18, 19], smoking [20], alcohol [21, 22], cardiovascular disease [23, 24], hypertension [25], mixed NCDs [26–29] and models of care [30, 31] in specific settings. However, to our knowledge, there has not been an attempt to collate information focussing on obesity in the same way.

The objective of this review is to explore the prevalence and incidence of overweight and obesity, and the changes in adiposity over time in populations directly affected by humanitarian crises; the cascade of care in these populations and perceptions of patients with overweight and obesity.

## Methods

A systematic review was conducted following the Preferred Reporting Items for Systematic Reviews and Meta-Analyses (PRISMA) 2020 guidelines [32] and applying the Synthesis Without Meta-analysis (SWiM) extension [33]. A scoping exercise was carried out in August 2019. This informed decisions about eligibility, inclusion dates and synthesis.

### Eligibility criteria

The PECO criteria described below form part of the eligibility criteria. Further, all study types, published in any language and carried out in any geographical location were considered eligible. For the scoping exercise, studies published from January 1st, 1999, were included. Reviewing the returns showed that the data being presented in the earlier papers were out of date given the context of changing levels of obesity globally. Since we were interested in providing a description which could be used by service providers in the current time, we restricted this review to papers published from January 1st, 2011, onwards.

Conference proceedings, letters, theses, clinical guidelines, opinion pieces and study protocols were excluded. Reports from NGOs are important in this field and were included as long as there was a description of the methods used to gather data.

### PECO criteria

The population, exposure, comparator and outcome (PECO) criteria for the study are described below.

**Population.** The population of interest was non-pregnant, civilian adults (aged 18 years or older) who had direct experience of humanitarian crises whether they were displaced or not. Economic migrants, Special Immigrant Visa entrants (those granted permanent residence in the USA for reasons including aiding US efforts in Afghanistan and Iraq [34]) and migrants unaffected by crises were not considered eligible. Service and military personnel, local combatants and prisoners of war were excluded. Service users attending general clinics were considered eligible, unless selected on the basis of a specific disease, when they were excluded. Studies with a mixed population were included if the population of interest could be clearly differentiated. For qualitative studies, this meant that the views of participants with overweight or obesity had to be identifiable. The study authors' definition of the type of migrant was applied.

**Exposure.** The crises of interest were armed conflict, complex emergencies and natural disasters (including earthquakes, landslides, tidal waves, tsunamis, floods, cyclones, hurricane and drought). Study authors' definitions of crises were applied. Exposures that began after or were ongoing in January 1999 were considered eligible. Exposures needed to be ongoing or previous to the time of data collection to be eligible. We did not impose other temporal restrictions on the exposure- outcome relationship.

We did not specifically search for COVID-19 related publications. We felt that the global nature of the pandemic meant that doing so would effectively result in a global prevalence estimate for overweight and obesity.

**Comparator.** Comparators were not considered as an eligibility criterium.

**Outcome.** Study authors' definitions of overweight and obesity were applied regardless of the measure and cut-offs used. During risk of bias (ROB) assessment the decisions made by authors in this regard were evaluated.

The primary outcomes of interest were:

1. The prevalence and incidence of overweight and / or obesity as defined by body mass index (BMI).

2. Change in adiposity over time in those diagnosed with overweight or obesity.

3. Cascade of care for overweight and / or obesity including recognition, seeking treatment or support and receiving treatment or support.

4. Patient knowledge and attitude to overweight and / or obesity.

   Secondary outcomes were:

1. Understanding of whether or not weight management is included as part of a wider programme of prevention or health promotion.

2. Barriers and facilitators to accessing treatment.

3. Evidence of use of other measures of adiposity.

## Information sources

Medline, Embase, PsycINFO, Cumulative Index of Nursing and Allied Health Literature (CINAHL) and Web of Science were searched. Grey literature and newly published peer reviewed material was identified by searching Google, ReliefWEB, UN High Commissioner for Refugees, WHO Institutional Repository for Information Sharing, UNICEF, Médecins Sans Frontières, International Rescue Committee, International Committee of the Red Cross, Centre for Disease Control and Prevention and Active Learning Network for Accountability and Performance (ALNAP). Search terms were adapted from our previous work [25] and can be seen in full in S1 Appendix. Searches were updated in January 2021 (databases) and May 2021 (Google searches). Rayyan was used to manage search returns [35].

## Selection processes

Two reviewers independently screened the titles and abstracts against the criteria described above. Conflicts were resolved by discussion. Papers included in the full text screening were also independently screened by two reviewers. Conflicts were again resolved by discussion. Reasons for exclusion were documented.

## Data collection

Data collection was carried out by one reviewer and independently checked by a second. Data were extracted into a shared spreadsheet. For each report, details of the publication (authors, year, title), study type, geographical context, a description of the population and a description of the exposure were extracted. For quantitative studies, method(s) of measurement, number with overweight and / or obesity, prevalence, sample size, measure of spread, details of subgroups and secondary outcomes were collected. For case control studies, we collected data from both cases and controls, but have presented data from controls only since cases may be systematically different from the general population due to the disease under study. For longitudinal studies data from each time point were extracted. In any study type, where subgroup data were available, these were extracted but only whole study level data are presented. For studies including adults and children, only data for those aged over 18 years was extracted. Where data for the whole study population were not presented, we used subgroup data to calculate these. In most cases this was done either by summing the numbers in mutually exclusive

subgroups, or by applying rates reported in subgroups to the population of the subgroup to give the number in each subgroup.

For qualitative studies (had any been identified) we planned to extract concepts, themes, barriers and facilitators as described by participants.

## Risk of bias assessment

A tool for risk of bias (ROB) in prevalence studies proposed by Hoy *et al* [36] was adapted for our study. The original tool included a question about the study population in relation to the national population. This was not appropriate for our study since we were not seeking nationally representative prevalence estimates. External ROB was judged on choice of sampling frame, method of sample selection and extent of non-responsiveness. Internal ROB was judged on method of data collection, case definition, choice of measure of adiposity, use of standardised procedures and accuracy of reporting. To add further granularity to the discussion about ROB, a score of low, medium or high risk was given in the external, internal and overall domains.

Had we identified any qualitative studies, we planned to use the Critical Appraisal Skills Programme (CASP) checklist [37].

ROB assessment was independently carried out by one reviewer and checked by a second. Discrepancies were resolved by discussion.

## Methods of synthesis

Our scoping exercise, initial search results and previous work in this field demonstrated that included studies were heterogeneous [25]. As a result, a narrative synthesis was carried out.

Definitions of overweight and obesity vary according to the anthropometric measure used and even with a single measure such as BMI, different cut-offs are proposed for different populations [38]. For the purposes of this review, we focussed on BMI as our scoping exercise showed that this was the most commonly used measure of adiposity amongst the included studies. Findings of overweight and obesity are reported as defined by individual study authors, but details of the cut-offs used were extracted for the ROB assessment.

Subgroups of interest for the synthesis included geographical setting, type of exposure, displacement status and ROB. Age and sex were considered important factors for the distribution of obesity and overweight [39]. Data are reported from all the included studies, but priority in interpretation is given to those with lower internal ROB since these studies are measuring the same phenomenon across the dataset. Heterogeneity was explored by describing the study type, population, exposure and setting of each study.

Data are presented in separate tables for high income countries (HICs) and low and middle income countries (LMICs), grouped by exposure type and location of study and shaded to indicate ROB. Categorisation as HIC or LMIC was selected to allow comparison to other publications in this field and to allow a rough assessment of resources available at a country level. The World Bank income-based classification system was used [40].

## Results

Overall, 20,376 non-duplicate search returns were identified and screened. Four hundred and eighty-one full-text reports were assessed for eligibility. Fifty-six reports from 45 studies were included in the review. Fifty reports were excluded because anthropometric measurements were presented in a way which did not allow categorisation and a further 13 were excluded as details were only given for underweight. The PRISMA flow diagram can be seen in Fig 1 and

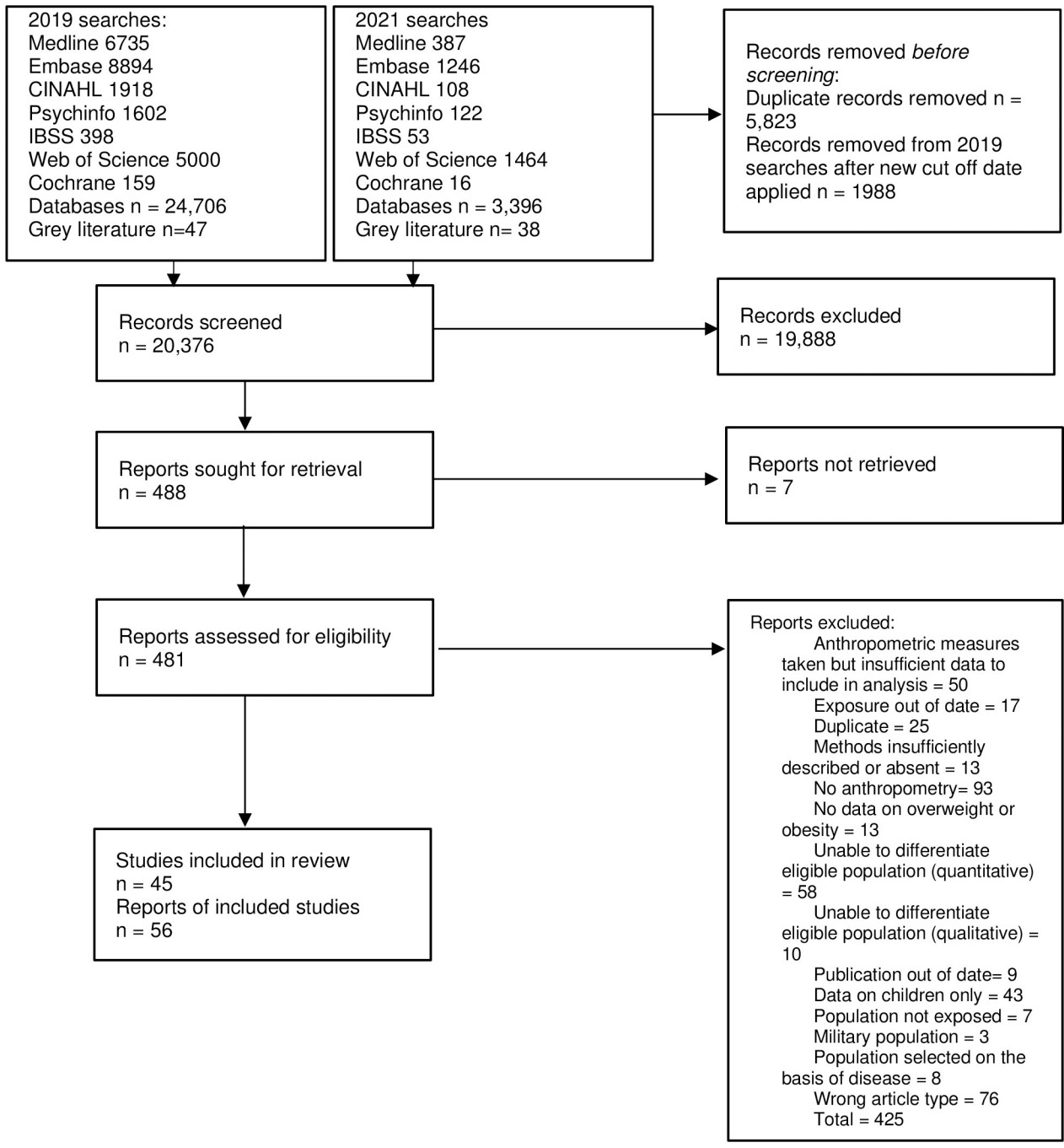

**Fig 1. Showing the PRISMA flow diagram.** *From*: Page MJ, McKenzie JE, Bossuyt PM, Boutron I, Hoffmann TC, Mulrow CD, et al. The PRISMA 2020 statement: an updated guideline for reporting systematic reviews. BMJ 2021;372:n71. doi: 10.1136/bmj.n71.

the S1 Checklist is included in the supplementary material. Reports excluded after full-text screening and the reasons for exclusion are detailed in S2 Appendix.

## Description of the included studies

Characteristics of the included studies are shown in Table 1.

**Table 1. Showing characteristics of included studies.**

| First author and Year | Title | Population and location of study | Description of crisis | Study design | Age included in data extraction |
|---|---|---|---|---|---|
| | | | **Conflict** | | |
| Yanni *et al* 2013 [42] | The health profile and chronic diseases comorbidities of US-bound Iraqi refugees screened by the International Organization for Migration in Jordan: 2007–2009 | Iraqi refugees in Jordan who are undergoing screening prior to entry to the USA | Not described, but most likely conflict | Results of routine health screening | 20 and older |
| Amr *et al* 2021 [55] | Mood and demographical factors as predictors of body mass index among Iraqi and Syrian refugees in Jordan | Iraqi and Syrian refugees living in Jordan | Iraqi and Syrian conflict | Cross sectional | 18 and older |
| Ratnayake *et al* 2020 [51] | Access to Care and Prevalence of Hypertension and Diabetes Among Syrian Refugees in Northern Jordan | Syrian refugees, non-pregnant and aged 18 or over, living outside of camps and located in Northern Jordan. Biological measurements, including height and weight, were taken from adults aged 30 or over | Syrian conflict | Cross sectional | 30 and older |
| Mansour *et al* 2020 [50] | Non-communicable diseases in Lebanon: results from World Health Organization STEPS survey 2017 | Syrian refugees and host Lebanese population residing in 8 governates in Lebanon | Syrian conflict | Cross sectional | 18–69 |
| Al-Duais and Al Awthan 2019 [54] | Association between qat chewing and dyslipidaemia among young males | Young men attending Ibb University, Yemen | War in Yemen | Cross sectional | 18–25 |
| Balcilar 2016 [57] | Health Status Survey of Syrian Refugees in Turkey | Syrian refugee population residing in 10 provinces of Turkey | Syrian conflict | Cross sectional | 18–69 |
| Eryurt and Menet 2020[52] | Noncommunicable diseases among Syrian refugees in Turkey: An emerging problem for a vulnerable group | Syrian refugees aged 18–69 living in Turkey | Syrian conflict | Cross sectional | 18–69 |
| Greene-Cramer *et al* 2020 [49] | Noncommunicable disease burden among conflict-affected adults in Ukraine: A cross-sectional study of prevalence, risk factors, and effect of conflict on severity of disease and access to care | Adult IDPs throughout Ukraine (excluding Donetsk & Luhansk regions) and adults (aged 30 or older) living in government-controlled, conflict-affected regions in Donbas regions in eastern Ukraine | Conflict in Ukraine starting in 2014 | Cross sectional | 30 and older |
| Singh *et al* 2015 [56] | Nutrition among men and household food security in an internally displaced persons camp in Kenya | Male IDPs in a camp in Kenya | Ethnic violence | Cross sectional | 18 and older |
| Chandra *et al* 2019 [41] | Prevalence of chronic disease risk factors in 35- to 44-year-old humanitarian arrivals to New South Wales (NSW), Australia | Newly arrived refugees to Sydney, Australia | Mixed exposures, majority from Iraq and Syria | Cross sectional | 35–44 |
| Drummond *et al* 2011 [43] | Knowledge of Cardiovascular Risk Factors in West African Refugee Women Living in Western Australia | Refugee women from Liberia or Sierra Leone living in Australia | Not described, likely varied | Cross sectional | 20–67 |
| Renzaho *et al* 2014 [44] | Obesity, Type 2 Diabetes and High Blood Pressure Amongst Recently Arrived Sudanese Refugees in Queensland, Australia | Sudanese adult refugees living in Brisbane, Australia | Conflict and other exposures in Sudan | Cross sectional | 18–70 |
| Maldari *et al* 2019 [53] | The health status of newly arrived Syrian refugees at the Refugee Health Service, South Australia, 2016 | Newly arrived Syrian adults and children who attended the Refugee Health Service in South Australia from 1 Jan to 31 Dec 2016 | Syrian Conflict | Cross sectional | 18 and older |
| Reznar *et al* 2020 [45] | The Burden of Chronic Health Conditions among Iraqi Refugees in Michigan | Iraqi refugees, Michigan, USA | Conflict since 2003, Iraqis resettled in the USA | Cross sectional | 18 and older |

*(Continued)*

**Table 1.** (Continued)

| First author and Year | Title | Population and location of study | Description of crisis | Study design | Age included in data extraction |
|---|---|---|---|---|---|
| Sastre *et al* 2020 [46] | From the Democratic Republic of the Congo to North Carolina: An Examination of Chronic Disease Risk | Refugees from Democratic Republic of Congo to the USA. Data collection October 2017 to March 2018 | Mixed exposures, from Democratic Republic of Congo | Cross sectional study | 18 and older |
| Jen *et al* 2015 [47] | Pre- and Post-displacement Stressors and Body Weight Development in Iraqi Refugees in Michigan | Iraqi refugees settled in Michigan, USA | Mixed exposures, Iraqi refugees to Michigan. Data collected after arrival in Michigan and then re-collected one year later | Cohort study | 18–69 |
| Jen *et al* 2018 [48] | Sex differences and predictors of changes in body weight and on-communicable diseases in a random newly-arrived group of refugees followed for two years | Iraqi refugees settled in Michigan, USA | Mixed exposures, Iraqi refugees to Michigan. Data collected after arrival in Michigan and then re-collected one year later and at year 2 follow up | Cohort study | 18–69 |
| **Long-standing Refugee Situation** | | | | | |
| Bhatta *et al* 2014 [60] | Socio-demographic and dietary factors associated with excess body weight and abdominal obesity among resettled Bhutanese refugee women in Northeast Ohio, United States | Bhutanese refugee women living in Northern Ohio, USA | Political and ethnic persecution of Bhutanese refugees of Nepalese origin | Cross sectional | 18–65 |
| Bhatta *et al* 2015 [59] | Chronic Disease Burden Among Bhutanese Refugee Women Aged 18–65 Years Resettled in Northeast Ohio, United States, 2008–2011 | Bhutanese refugee women living in Northern Ohio, USA | Political and ethnic persecution of Bhutanese refugees of Nepalese origin | Cross sectional | 18–65 |
| Kumar *et al* 2014 [68] | Noninfectious disease among the Bhutanese refugee population at a United States urban clinic | Adult Bhutanese refugees who attended Grady Refugee clinic, Atlanta, Georgia, USA | Political and ethnic persecution of Bhutanese refugees of Nepalese origin | Cross sectional | 18 and older |
| Bayyari *et al* 2013 [58] | Dieting behaviours, obesity and predictors of dieting among female college students at Palestinian universities | Female Palestinian college students in Palestine | Long term Palestinian situation | Cross sectional | Young adults |
| El Kishawi *et al* 2014 [61] | Obesity and overweight: prevalence and associated socio demographic factors among mothers in three different areas in the Gaza Strip-Palestine: a cross-sectional study | Mothers in the Gaza Strip, Palestine | Long term Palestinian situation | Cross sectional | 18–50 |
| Kory *et al* 2013 [73] | Health ramifications of the Gush Katif evacuation | Residents evacuated from Gush Katif, Gaza Strip, Palestine | Long term Palestinian situation | Cross sectional data from a cohort | 21 and older |
| Dhair and Abed 2020 [62] | The association of types, intensities and frequencies of physical activity with primary infertility among females in Gaza Strip, Palestine: A case-control study | Couples of reproductive age living in the Gaza Strip, Palestine | Long term Palestinian situation | Case-control study | 18–49 |
| Damiri *et al* 2018 [70] | Metabolic syndrome among overweight and obese adults in Palestinian refugee camps | Palestinian refugees displaced due to conflict and who have lived in one of three camps in Nablus, West Bank, Palestine for at least 6 months | Long term Palestinian situation | Cross sectional | 28–65 |
| Damiri *et al* 2019 [71] | Metabolic syndrome and related risk factors among adults in the northern West Bank, a cross-sectional study | Palestinian adults aged living in the West Bank, Palestine | Long term Palestinian situation | Cross sectional | 18–70 |
| Abdollahi *et al* 2015 [67] | High occurrence of food insecurity among urban Afghan refugees in Pakdasht, Iran 2008: a cross-sectional study | Afghan refugees living in Pakdasht, Tehran, Iran | Conflict and other exposures in Afghanistan | Cross sectional | 24–60 |

*(Continued)*

**Table 1.** (Continued)

| First author and Year | Title | Population and location of study | Description of crisis | Study design | Age included in data extraction |
|---|---|---|---|---|---|
| Taherifard et al 2021 [69] | The prevalence of risk factors associated with non-communicable diseases in Afghan refugees in southern Iran: a cross-sectional study | Afghan refugees in Southern Iran refugee camp | Mixed exposure, refugees leaving Afghanistan since 1979 settling in Iran | Cross sectional | 25 and older |
| Naigaga et al 2018 [72] | Body size perceptions and preferences favor overweight in adult Saharawi refugees | Refugees from Western Sahara that have settled in the Algerian desert | Western Sahara War | Cross sectional | 18–80 |
| Kim et al 2015 (NORNS 2) [64] | Vitamin D status and associated metabolic risk factors among North Korean refugees in South Korea: a cross-sectional study | North Korean refugees resident in Seoul, South Korea. Part of the NORNS study | Long standing refugee situation, North Korean refugees | Cross sectional data from ongoing cohort study | 30–81 |
| Jung Kim et al 2016 (NORNS 3) [65] | Prevalence of metabolic syndrome and its related factors among North Korean refugees in South Korea: a cross-sectional study | | | | 30 and older |
| Kim et al 2018 (NORNS 1) [63] | Prevalence of general and central obesity and associated factors among north Korean refugees in South Korea by duration after defection from North Korea: A cross-sectional study | | | | 20–60 |
| Jeong et al 2017 (NORNS 4) [66] | Changes in body weight and food security of adult North Korean refugees living in South Korea | | | | 19 and older |
| **Natural disasters** | | | | | |
| Furusawa et al 2011 [74] | Communicable and non-communicable diseases in the Solomon Islands villages during recovery from a massive earthquake in April 2007 | Communities affected by the earthquake in the Solomon Islands | Earthquake leading to tsunami and landslides | Cross sectional | 18 and older |
| Herrera-Fontana et al 2020 [75] | Food insecurity and malnutrition in vulnerable households with children under 5 years on the Ecuadorian coast: a post-earthquake analysis | Households located in La Punta, a rural community, located 40 minutes by road from the epicentre of the Ecuadorian earthquake on 16 April 2016. Data on overweight and obesity is reported for 'mothers or women responsible for the household' | An earthquake of magnitude 7.8 on the Richter scale in the province of Manabi on the Ecuadorian coast | Cross sectional | 18–60 |
| Adrega et al 2018 [76] | Prevalence of cardiovascular disease risk factors, health behaviours and atrial fibrillation in a Nepalese post-seismic population: A cross-sectional screening during a humanitarian medical mission | Inhabitants of 14 villages in Sindhupalchok, a northern region of Nepal, located in the epicentre of the earthquake | May 2015 earthquake in Nepal | Cross sectional | 18 and older |
| Sakai et al 2020 (FHMS 1) [77] | Relationship between the prevalence of polycythemia and factors observed in the mental health and lifestyle survey after the Great East Japan Earthquake | Those forced to evacuate due to the Great East Japan Earthquake. Subset of participants from the Fukushima Health Management Survey | Great East Japan Earthquake, 2011 | Cross sectional | 20–90 |
| Ohira et al 2016 (FHMS 2) [78] | Effect of evacuation on body weight after the Great East Japan Earthquake | Japanese men and women living in communities near the Fukushima Daiichi Nuclear Power Plant in the Fukushima prefecture, Japan. Subset of participants from the Fukushima Health Management Survey | Great East Japan Earthquake, 2011 | Cohort study | 40–90 |
| Ohira et al 2017 (FHMS 3) [79] | Changes in Cardiovascular Risk Factors After the Great East Japan Earthquake: A Review of the Comprehensive Health Check in the Fukushima Health Management Survey | Residents living near the Fukushima Daiichi power plant, Japan. Subset of participants from the Fukushima Health Management Survey | Great East Japan Earthquake, 2011 | Cross sectional | 40–90 |

*(Continued)*

**Table 1.** (*Continued*)

| First author and Year | Title | Population and location of study | Description of crisis | Study design | Age included in data extraction |
|---|---|---|---|---|---|
| Satoh *et al* 2021 (FHMS 4) [80] | Relationship between risk of hyper-low-density lipoprotein cholesterolemia and evacuation after the Great East Japan Earthquake | Japanese adults with health insurance without a diagnosis of hyper-LDL cholesterolemia living near the Fukushima Daiichi nuclear power plant, Japan. Subset of participants from the Fukushima Health Management Survey | Great East Japan Earthquake, 2011 | Prospective cohort | 40–89 |
| Takahashi *et al* 2016 [81] | Weight Gain in Survivors Living in Temporary Housing in the Tsunami-Stricken Area during the Recovery Phase following the Great East Japan Earthquake and Tsunami | Research project for prospective Investigation of health problems Among Survivors of the Great East Japan Earthquake and Tsunami Disaster—The survey was carried out between September 2011 and February 2012 in 3 municipalities in Iwate Prefecture located in the Tohoku area in the northern part of Honshu, Japan | Great East Japan Earthquake, 2011 | Prospective cohort study | 18 and older |
| Takahashi *et al* 2020 [82] | Increased incidence of metabolic syndrome among older survivors relocated to temporary housing after the 2011 Great East Japan earthquake & tsunami | | | | |
| Takahashi *et al* 2021 [83] | Increase in Body Weight Following Residential Displacement: 5-year Follow-up After the 2011 Great East Japan Earthquake and Tsunami | | | | |
| Hikichi *et al* 2019 [86] | Residential relocation and obesity after a natural disaster: A natural experiment from the 2011 Japan Earthquake and Tsunami | Survivors of 2011 earthquake and tsunami living in Iwanuma, Japan | Great East Japan Earthquake, 2011 | Natural experiment nested within cohort study | 66 and older |
| Ebner *et al* 2016 [84] | Lifestyle-related diseases following the evacuation after the Fukushima Daiichi nuclear power plant accident: a retrospective study of Kawauchi Village with long-term follow-up | Residents who attended National Health Screening programmes, Kawauchi village, Japan | Great East Japan Earthquake, 2011 | Cohort study | 40 and older |
| Nakamura *et al* [85] | Psychological distress as a risk factor for dementia after the 2004 Niigata-Chuetsu earthquake in Japan | Residents living in Ojiya city, Japan, who did the annual health check examination after the 2004 earthquake | 2004 Niigata-Chuetsu earthquake | Cohort study | 40 and older |
| | | | **Mixed exposures** | | |
| Mulugeta *et al* 2018 [87] | Longitudinal Changes and High-Risk Subgroups for Obesity and Overweight/Obesity Among Refugees in Buffalo, NY, 2004–2014 | Adult and child refugees attending the Jericho Road Community Health Centre, Buffalo, New York, USA | Service open to all refugees | Retrospective cohort | 19 and older |
| Mulugeta *et al* 2019 [88] | Burden of Mental Illness and Non-communicable Diseases and Risk Factors for Mental Illness Among Refugees in Buffalo, NY, 2004–2014 | Adult refugees attending the Jericho Road Community Health Centre, Buffalo, New York, USA | Service open to all refugees | Cross sectional | 18 and older |
| Mulugeta *et al* 2019 [89] | Disease Burdens and Risk Factors for Diabetes, Hypertension, and Hyperlipidemia among Refugees in Buffalo, New York, 2004–2014 | Adult refugees attending the Jericho Road Community Health Centre, Buffalo, New York, USA | Service open to all refugees | Cross sectional | 18 and older |
| Rhodes *et al* 2016 [90] | Development of Obesity and Related Diseases in African Refugees After Resettlement to United States | African refugees resettled to Rhode Island, USA, 2004–2007. Excluded children, pregnant women, and individuals without an electronic medical record or recorded height | Mixed exposures, majority were African refugees (Eritrea, Ethiopia, Ghana, Liberia, Rwanda and Somalia) | Cohort Study | 18 and older |
| Bardenheier *et al* 2019 (1) [91] | Prevalence of tuberculosis disease among adult US-bound refugees with chronic kidney disease | Medical examination of all adult refugees before arriving in the USA | Various exposures as refugees from multiple locations | Cross sectional | 18 and older |

(*Continued*)

**Table 1.** (Continued)

| First author and Year | Title | Population and location of study | Description of crisis | Study design | Age included in data extraction |
|---|---|---|---|---|---|
| Bardenheier *et al* 2019 (2) [92] | Trends in Chronic Diseases Reported by Refugees Originating from Burma Resettling to the United States from Camps Versus Urban Areas During 2009–2006 | Medical examination of all Burmese refugees before arriving in the USA | Violence, conflict and natural disasters | Cross sectional | 18 and older |
| Nguyen and Rehkopf, 2016 [93] | Prevalence of Chronic Disease and Their Risk Factors Among Iranian, Ukrainian, Vietnamese Refugees in California, 2002–2011 | The three refugee populations with the greatest number of arrivals to California, USA, between 1995 and 2011 | Mixed exposures, majority from Iran, Ukraine and Vietnam | Cross sectional | 18 and older |
| Amstutz *et al* 2020 [94] | Nutritional Status and Obstacles to Healthy Eating Among Refugees in Geneva | All non pregnant adults in the asylum process in Geneva, Switzerland, collected between June 2017 and March 2019 | Mixed exposures, majority from Afghanistan, Eritrea, Sri Lanka and Syria | Cross sectional, mixed methods | Adults |
| Modesti *et al* 2020 [95] | Blood pressure and fasting glucose changes in male migrants waiting for an asylum decision in Italy. A pilot study | Male asylum seekers waiting for an asylum decision in Italy for between 0 and 30 months | Mixed exposures, majority of asylum seekers from Eritrea, Ghana, Guinea and Nigeria | Cross sectional | 18 to 40 |
| Kortas *et al* 2017 [96] | Screening for infectious diseases among asylum seekers newly arrived in Germany in 2015: a systematic single-centre analysis | Asylum seekers at reception centre in Ausberg, Germany in 2015 | Various exposures as refugees from multiple locations. Mainly Afghanistan, Albania, Eritrea, Syria and Nigeria | Cross sectional | 18 to 75 |

USA = United States of America

STEPS = WHO STEPwise approach to surveillance of noncommunicable diseases

IDPs = internally displaced people/ persons

LDL = low density lipoprotein

NY = New York

NORNS = North Korean Refugee Health in South Korea study

In terms of the exposure, seventeen reports related to conflict situations [41–57], sixteen to long-standing refugee situations [58–73], thirteen to natural disasters [74–86] and ten included mixed exposures [87–96]. Several crises were the subject of multiple studies. The Great East Japan Earthquake was the exclusive exposure of nine reports [77–84, 86], the internal conflict in Syria of seven reports [41, 50–53, 55, 57] and the Palestinian situation of six reports [58, 61, 62, 70, 71, 73]. The map in Fig 2A shows the countries where exposures occurred, based on the number of reports mentioning those countries. The most frequently examined exposures were in Japan [77–86] (10 reports), followed by Syria [41, 50–53, 55, 57, 94, 96] (9 reports), then Palestine [58, 61, 62, 70, 71, 73] (6 reports) and Iraq [41, 42, 45, 47, 48, 55] (6 reports).

With regards to setting, thirty five reports were from studies carried out in HICs, twenty were conducted in LMICs and one was conducted in both settings [91]. The map in Fig 2B shows the frequency of reports from different countries. The most common study countries were the United States [45–48, 59, 60, 68, 87–90, 93] (12 reports), then Japan [77–86] (10 reports) followed by Palestine [58, 61, 62, 70, 71, 73] (6 reports).

Data reported in this paper were all collected after the exposure had begun. For those in long-standing refugee situations or going through asylum seeking or refugee resettlement processes, the exposure was considered to be ongoing. For those exposed to natural disasters, data collection took place between 4 months [75] and 4 years after the disaster [80, 82].

Forty reports considered displaced populations, while four considered non-displaced populations [54, 58, 75, 84] and ten included both displaced and non-displaced participants [49, 61,

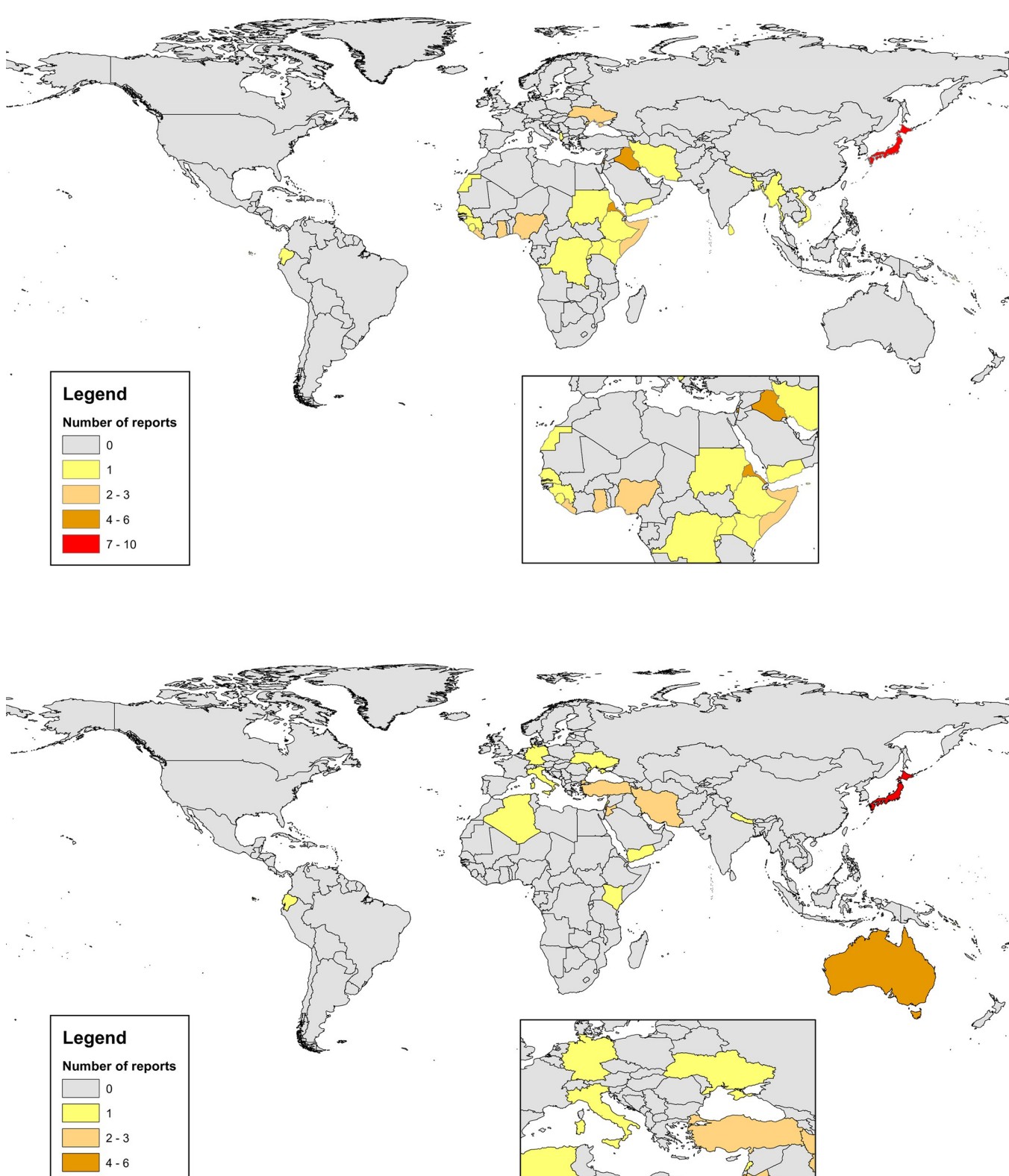

**Fig 2.** A. Showing the frequency of reports by country of exposure. B Showing frequency of reports by study location. Maps were produced in ArcGIS from ESRI using base map data from the World Food Programme accessed via The National Archives (UK). Contains public sector information licensed under the Open Government Licence v3.0.

62, 74, 78–83]. Displacement status was unclear for two reports [71, 85]. Of the HIC reports, twenty-six were related to populations who had been displaced from LMICs.

The sample size also varied across the studies with some reports including as many as 444356 participants [91] whilst the smallest study only included 28 participants [75]. Only 8 studies reported measures of spread for either population level or subgroup estimates [44, 50, 51, 57, 62, 81, 82, 84].

Although studies may have included children, we have restricted data extraction to adults only and the age range of adults included is seen in Table 1. Most studies sampled a wide range of ages, but some were restricted to young adults [54, 58], middle aged adults [41] or older adults [86]. Most studies included men and women. However, Al-Duais and Al-Awthan [54], Modesti *et al* [95] and Singh *et al* [56] included men only. Bhatta *et al* [59, 60], Bayyari *et al* [58], El Kishawi *et al* [61], Herrera-Fontana *et al* [75], Drummond *et al* [43] and Dhair and Abed [62] included females only.

## Prevalence of overweight and obesity

Tables 2 and 3 show the prevalence of overweight, obesity and overweight or obesity combined for LICs and HICs, respectively. Forty-seven reports used WHO recommended BMI cut-offs of 25 kg/m$^2$ and 30 kg/m$^2$ to define overweight and obesity, respectively [97] or regional variations. Nine reports used non-standard definitions where a single cut-off was applied for both overweight and obesity, or no justification was given for the choice of cut-off used [43, 45, 65, 76, 77, 80, 81, 83, 85].

Prevalence was reported for the whole populations or by subgroups according to the aim of the study. As can be seen by entries in bold in Tables 2 and 3, where information was not available for the whole population, it could be calculated.

In whole populations, prevalence rates for overweight and obesity combined ranged from 6.4% [56] to 82.8% [51]. For overweight alone they ranged from 6.0% [56] to 53% [43] and for obesity alone from 0% [54] to 52.7% [51].

## Prevalence of overweight and obesity in subgroups

The prevalence ranges for our subgroups of interest are presented in Fig 3, derived from data presented in Appendix Three.

The range of prevalence estimates remains wide across overweight, obesity and the two combined in most of the subgroups, however, it appears the widest for the estimates of overweight and obesity combined. The upper and lower bounds of the obesity estimates are generally lower than those of the overweight estimates.

In terms of exposures, conflict generates the widest range of estimates. The mixed exposures group generates the narrowest range of estimates across all three metrics. This group includes studies which were carried out as part of routine pre-immigration or post-immigration health checks.

The lowest internal ROB and being displaced are associated with the widest range of prevalence estimates for their subgroups.

It is interesting to note that prevalence estimates are similar for studies carried out in both HICs and LMICs. In studies with low internal ROB, estimates for overweight and obesity combined were between 6.4% [56] and 82.8% [51] for LMICs and between 15.9% [95] and 65% [41] for HICs. Taking a closer look at geography, reports from the NORNS study [63–66] report relatively homogeneous results (overweight 22% [63] to 23.4% [64], obesity 18.9% [64] to 20.34% [65] and the two combined 42.1% [63] to 43% [66]), whereas those from earthquake afflicted populations in Japan are varied (overweight 23.2% [86] to 29.4% [78], obesity 3.4%

**Table 2. Showing prevalence of overweight and obesity in studies carried out in low and middle income countries.**

| First Author, Year | Number (proportion %) with overweight | Number (proportion %) with obesity | Number (proportion %) with overweight or obesity | Value of denominator | Country | BMI cut-offs applied | Displaced |
|---|---|---|---|---|---|---|---|
| | | | **Conflict** | | | | |
| Yanni et al 2013 [42] | 4495 (38) | 3982 (34) | 8477 (71) | 11898 | Jordan | Standard | Yes |
| Amr et al 2019 [55] | NR | NR | 628 (60.5) | 1038 | Jordan | Standard | Yes |
| Ratnayake et al 2020 [51] | 273 (30.1) | 487 (52.7) | 751 (82.8) | 907 | Jordan | Standard | Yes |
| Mansour et al 2020 [50] | 661 (34.2) | 573 (28.6) | 1234 (57.8) | 2134 | Lebanon | Standard | Yes |
| Al Duais and Al Awthan 2019 [54] ^ | 17 (22.1) | 0 | 17 (22) | 77 | Yemen | Standard | No |
| Balcilar 2016 [57] | 1762 (32.6) | 1498 (27.7) | 3260 (60.3) | 5407 | Turkey | Standard | Yes |
| Eryurt and Menet 2020 [52] | NR | NR | 3503 (64) | 5492 | Turkey | Standard | Yes |
| Greene-Cramer et al 2020 [49] | NR | NR | 241 (17.2) | 1400 | Ukraine | Standard | Mixed |
| Singh et al 2015 [56]** | 15 (6) | 1 (0.4) | 16 (6.4) | 251 | Kenya | Standard | Yes |
| | | | **Long-standing Refugee Situation** | | | | |
| Bayyari et al 2013 [58] * | 51 (12.4) | 7 (1.7) | 58 (14.1) | 410 | Palestine | Standard | No |
| El Kishawi et al 2014 [61] * | 123 (34.5) | 105 (29.7) | 228 (63.9) | 357 | Palestine | Standard | Mixed |
| Kory et al 2013 [73] | 166 (31.6) | 93 (17.7) | 259 (49.3) | 525 | Palestine | Standard | Yes |
| Dhair and Abed 2020 [62] ^^ | 61 (38.1) | 36 (22.5) | 97 (60.6) | 160 | Palestine | Standard | Mixed |
| Damiri et al 2018 [70] | 188 (27.3) | 246 (35.7) | 435 (63.1) | 689 | Palestine | Standard | Yes |
| Damiri et al 2019 [71] | 348 (32.2) | 345 (33.1) | 693 (64.1) | 1082 | Palestine | Standard | Unclear |
| Abdollahi et al 2015 [67] | 440 (37.3) | 239 (20.3) | 769 (57.6) | 1178 | Iran | Standard | Yes |
| Taherifard et al 2021 [69] | 58 (28.0) | 46 (22.2) | 104 (50.2) | 207 | Iran | Standard | Yes |
| Naigaga et al 2018 [72] | 91 (27.6) | 49 (14.8) | 140 (42) | 330 | Algeria | Standard | Yes |
| | | | **Natural disasters** | | | | |
| Furusawa et al 2011 [74] | 122 (45.7) | 41 (15.4) | 163 (61.0) | 267 | Solomon Islands | Standard | Mixed |
| Herrera-Fontana et al 2020 [75]* | 10 (35.7) | 6 (21.4) | 16 (57.1) | 28 | Ecuador | Standard | No |
| Adrega et al 2018 [76] + | NR | NR | 41 (25) | 164 | Nepal | non-standard | Yes |

NR = not reported and can't be calculated from available information

Standard = WHO definition of overweight and obesity used.

Regional = regional cut-offs as defined by national guidelines for overweight and obesity used

Non-standard = neither WHO nor regionally defined cut-offs used.

Green = low internal ROB

Orange = moderate internal ROB

No Fill = high internal ROB

* = females only

** = males only

^ = males only. Data for non-Qat chewers presented

^^ = Females only. Data for controls presented

+ = reported as any BMI > = 25

[78] to 32.7% [83] and the two combined 26.4% [85] to 81.6% [81]). Several studies examined populations from the Middle East region [42, 45, 47, 48, 50–55, 57, 58, 61, 62, 67, 69–71, 73]. The range of estimates for overweight (12.4% [58] to 38.3% [47]), obesity (0% [54] to 52.7% [51] and the two combined (14.1% [58] to 82.8% [51]) remain wide. Nguyen *et al* compared

**Table 3. Showing prevalence of overweight and obesity in studies carried out in high income countries.**

| First Author, Year | Number (proportion %) with overweight | Number (proportion %) with obesity | Number (proportion %) with overweight or obesity | Value of denominator | Country of study | BMI cut-offs applied | Displaced |
|---|---|---|---|---|---|---|---|
| **Conflict** | | | | | | | |
| Chandra et al 2019 [41] | **107 (45)** | **66 (28)** | 173 (65) | 237 | Australia | Standard | Yes |
| Drummond et al 2011 [43] * | **27 (53)** | **14 (27)** | **41 (80)** | 51 | Australia | non-standard | Yes |
| Renzaho et al 2014 [44] | 97 (30.9) | 63 (20.1) | **160 (51)** | 314 | Australia | Standard | Yes |
| Maldari et al 2019 [53] | 48 **(25.8)** | 82 **(44.1)** | 130 **(69.9)** | 186 | Australia | Standard | Yes |
| Reznar et al 2020 [45] | NR | NR | **403 (65.7)** | 613 | USA | non-standard | Yes |
| Sastre et al 2020 [46] | 17 (37) | 12 (26) | **29 (63)** | 48 | USA | Standard | Yes |
| Jen et al 2015 [47] | 111 (38.3) | 56 (19.3) | **167 (57.6)** | 298 | USA | Standard | Yes |
| Jen et al 2018 [48] + | NR | NR | NR | 282 | USA | Standard | Yes |
| **Long-standing Refugee Situation** | | | | | | | |
| Bhatta et al 2014 [60] * | NR | NR | **70 (64.8)** | 108 | USA | Regional | Yes |
| Bhatta et al 2015 [5] * | 38 (35.2) | 32 (29.6) | **70 (64.8)** | 108 | USA | Regional | Yes |
| Kumar et al 2014 [68] | 28 (42) | 6 (9) | 34 (52) | 66 | USA | Standard | Yes |
| Kim et al (NORNS 2) 2015 [64] | 89 **(23.4)** | 72 **(18.9)** | **161 (42.3)** | 381 | South Korea | Regional | Yes |
| Kim et al (NORNS 3) 2016 [65] | NR | **144 (20.34)** | NR | 708 | South Korea | non-standard | Yes |
| Kim et al 2018 (NORNS 1) [63] | **203 (22)** | **183 (20)** | **386 (42.1)** | 917 (BMI) | South Korea | Regional | Yes |
| Jeong et al (NORNS 4) 2017 [66] | 34 (22.8) | 30 (20.1) | **64 (43)** | 149 | South Korea | Regional | Yes |
| **Natural disasters** | | | | | | | |
| Sakai et al FHMS1 2020 [77] | (NR) | 9718 (33.2) | (NR) | 29267 | Japan | non-standard | Yes |
| Ohira et al FHMS2 2016 [78] | **8070 (29.4)** | **928 (3.4)** | **8998 (32.7)** | 27486 | Japan | Standard | Mixed |
| Ohira et al FHMS3 2017 [79] | NR | NR | NR | 27486 | Japan | Standard | Mixed |
| Satoh et al FHMS4 2021 [80] | NR | NR | **5359 (28.7)** | 18670 | Japan | non-standard | Mixed |
| Takahashi et al (2) 2016 [81]++ | NR | NR | **2122 (81.6)** | 6601 | Japan | non-standard | Mixed |
| Takahashi et al (3) 2020 [82] | NR | 1544 **(21.1)** | NR | 7318 | Japan | Standard | Mixed |
| Takahashi et al (1) 2021 [83] | NR | **3238 (32.7)** | NR | 9897 | Japan | non-standard | Mixed |
| Hikichi et al 2019 [86] | **827 (23.2)** | **967 (27.1)** | **1794 (50.3)** | 3567 | Japan | Standard | Yes |
| Ebner et al 2016 [84] | NR | **277 (35.3)** | NR | **784** | Japan | Regional | No |
| Nakamura et al 2019 [85]++ | NR | NR | 1496 **(26.4)** | 5674 | Japan | non-standard | Unclear |
| **Mixed** | | | | | | | |

*(Continued)*

**Table 3.** (*Continued*)

| First Author, Year | Number (proportion %) with overweight | Number (proportion %) with obesity | Number (proportion %) with overweight or obesity | Value of denominator | Country of study | BMI cut-offs applied | Displaced |
|---|---|---|---|---|---|---|---|
| Mulugeta *et al* 2018 (1) [87] | **345 (32.7)** | **158 (15)** | **503 (47.7)** | 1055 | USA | Standard | Yes |
| Mulugeta *et al* 2019 (2) [88] | **345 (32.7)** | **158 (15)** | 503 (47.7) | 1055 | USA | Standard | Yes |
| Mulugeta *et al* 2019 (3) [89] | NR | 361 (23) | NR | 1570 | USA | Standard | Yes |
| Rhodes *et al* 2016 [90]^ | NR | 17 (13.5) | NR | 126 | USA | Standard | Yes |
| Bardenheier *et al* 2019 (1) [91] | 120993 (27.2) | 85231 (19.2) | **206224 (46.4)** | 444356 | (USA) ^^ | Standard | Yes |
| Bardenheier *et al* 2019 (2) [92] | **22136 (30.2)** | **7342 (10)** | **29478 (40.2)** | 73251 | (USA) ^^^ | Standard and regional | Yes |
| Nguyen and Rehkopf, 2016 [93] | NR | 4457 **(20.3)** | NR | 21968 | USA | Standard | Yes |
| Amstutz *et al* 2020 [94] | 110 (31.2) | 38 (10.8) | **148** (42) | 352 | Switzerland | Standard | Yes |
| Modesti *et al* 2020 [95] ** | 30 **(15.4)** | 1 **(0.5)** | 31 **(15.9)** | 195 | Italy | Standard | Yes |
| Kortas *et al* 2017 [96] | **295 (9.3)** | **116 (23.7)** | **411 (33)** | **1246** | Germany | Standard | Yes |

NR = not reported and can't be calculated from available information

Standard = WHO definition of overweight and obesity used.

Regional = regional cut-offs as defined by national guidelines for overweight and obesity used

Non-standard = neither WHO nor regionally defined cut-offs used.

Green = low internal ROB

Orange = moderate internal ROB

No fill = high internal ROB

* = Females only

** = Males only

+ = results presented graphically with insufficient information to extract data

++ = reported as any BMI > = 25

^ = presenting data for refugees only

^^ = included US bound refugees from a wide range of countries

^^^ = included US bound Burmese refugees from a number of South East Asian countries

refugees who had relocated to California from Iraq, Vietnam and Ukraine. They report that those from Ukraine were more likely to be obese or severely obese than the other nationalities, (Adjusted Odds Ratio (AOR) 2.1; CI 1.9–2.3) and (AOR 2.5; CI 2.1–2.8) respectively [93].

Five reports had strikingly low estimates [54, 56, 58, 95, 96] with a combined prevalence of between 6.4% [56] and 23.7% [96].Three reports included male participants only [54, 56, 95], and one was majority (75.4%) male [96]. They also included younger participants either as part of their sampling strategy or due to attendees at the services. The report with the youngest mean age was Bayyari *et al*, at 20.1 (standard deviation (SD) 1.2) years [58] and the oldest was Singh *et al* at 37 (SD 16) years [56].

Three reports had strikingly high prevalence estimates [43, 51, 81]. They all sampled populations originating from and living in different geographical settings and with different exposures. Drummond *et al* [43] examined West African women only with a mean age of 35 (SD 10.6) years and found a combined prevalence of 80%. Ratnayake *et al* [51] and Takahashi *et al*

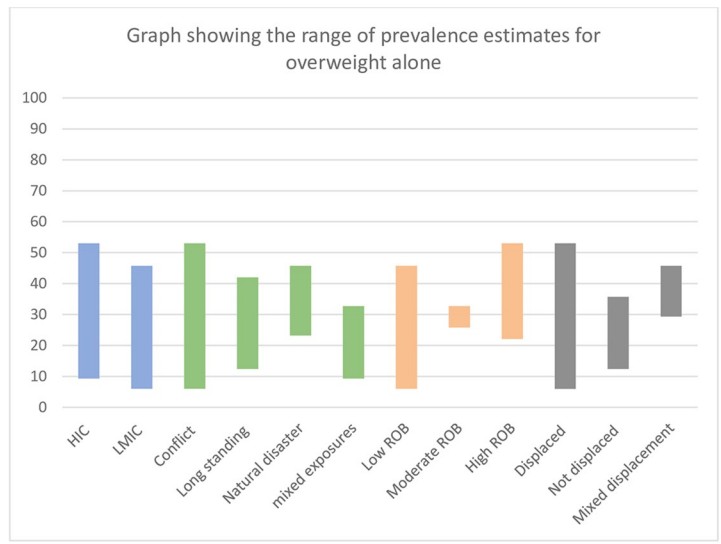

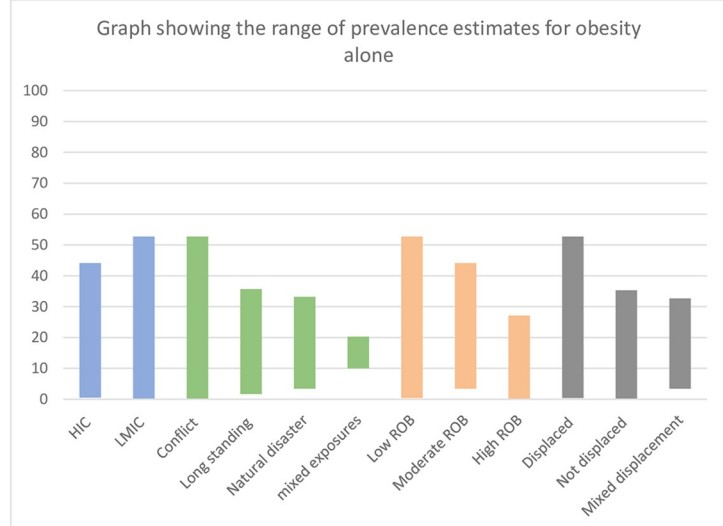

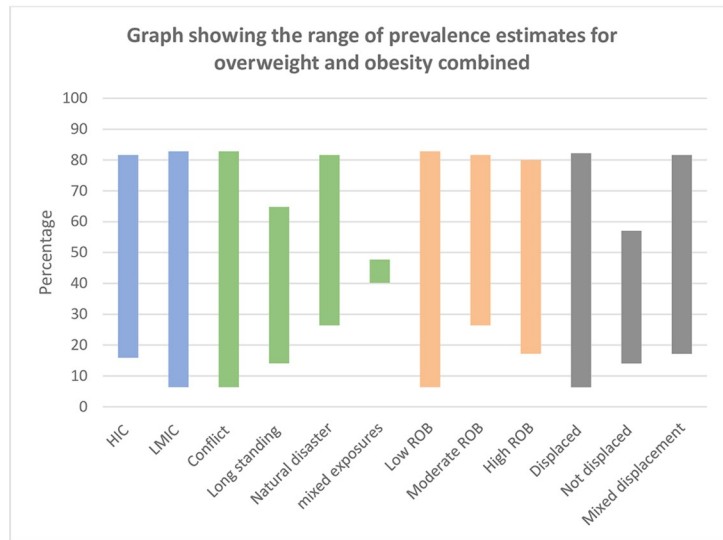

**Fig 3. Showing the range of prevalence estimates in subgroups of interest for overweight, obesity and the two combined.**

[81] sampled both sexes with an older mean age of 56 (SD 13,2) and 60 years for men, 64 years for women, respectively and found combined prevalences of 82.8% and 81.6% respectively.

Some reports did formally compare overweight and / or obesity prevalence estimates between men and women. Several found that these measures were higher in women compared to men [51, 52, 69, 71, 72, 93, 94]. However, Mansour *et al* found higher rates of obesity in women, but no difference in rates of overweight between the sexes [50]. Damiri *et al* and Balcilar also report that whilst obesity is more prevalent in women, overweight is more prevalent in men [57, 70].

All the studies that explored the relationship between age and adiposity found that prevalence estimates increased with age [45, 51, 52, 57, 59, 61, 69, 72, 93, 94].

## Changes over time and with displacement

Findings related to displacement and longitudinal changes are difficult to tease out, so are reported here together.

Whilst studies did not report on incidence specifically, ten reports mention change in prevalence of obesity and overweight over time. Nine [47, 48, 66, 78, 79, 84, 86, 87, 90] of these reports considered populations in HICs and only one was in a LMIC [73]. All dealt with displaced populations. Of the ten reports, seven noted an increase [47, 48, 78, 79, 86, 87, 90] and one [73] reported no change over time. Two reports suggested an initial increase followed by a decrease, stabilisation or loss of adiposity [66, 84].

Of those that reported increases in overweight or obesity, Jen *et al* found that in the first year following relocation to the United States there was a significant increase in BMI and an upward shift in the prevalence of overweight and obesity amongst refugee populations [47]. Mulugeta *et al* found that for every additional year refugees lived in the USA, the risk of overweight or obesity increased by 23% among men (Odds Ratio (OR) = 1.23; 95% CI = 1.09–1.39) and 18% among women (OR = 1.18; 95% CI = 1.04–1.35) when adjusted for confounders [87].

Takahashi et al contribute further to the importance of place of displacement. They report significant increases in body weight in people relocated to temporary housing compared to those not relocated over a five year observation period [83].

Considering changes in BMI over time without categorising into overweight and obesity, three reports noted increases in BMI [65, 66, 94]. However, Modesti *et al* found no strong evidence for an association between time in an Italian immigration centre and increase in BMI over a 30 month period [95].

Four reports formally compared changes in adiposity before and after exposure to the Great East Japan Earthquake [78, 79, 84, 86]. Hikichi et al report that approximately 2.5 years after the disaster, the prevalence of obesity had increased amongst those displaced (25.0% to 35.1%) but decreased amongst those not displaced (26.9% to 26.6%) compared to 7 months before the disaster. [86] Ebner *et al* report that the OR of obesity was higher in the year after the disaster, but that this risk was no longer significant in the second year after the disaster (OR 1.31 (95% CI 1.06 to 1.38) and 1.07 (95% CI 0.93 to 1.24) respectively) [84].

Ohira et al report that BMI and obesity increased in earthquake affected populations. This increase was greater in those evacuated compared to those not evacuated and greater in males compared to females [78, 79]. The multivariable adjusted hazard ratio for overweight after the disaster was 1.61 (95% CI 1.47 to 1.77) [78].

Only one non-earthquake study compared BMI before and after exposure. No change was found [73].

### Other outcomes

The other outcomes of interest were considered less frequently. There were no papers reporting on the cascade of care for obesity. However, attempts have been made to gather information about risk factors for higher BMI and targets for primary prevention. Balcilar, 2016 reports that 14.1% of Syrian refugees in Turkey were advised to reduce their fat intake [57]. Several reports from countries in the Middle East show that there are poor levels of fruit and vegetable intake and low levels of physical activity in refugee populations in general [50, 52, 57, 69].

We did not identify any qualitative studies which met our selection criteria. However, in a cross-sectional study measuring both self-perceived body size and BMI in Saharawi refugees, Naigaga *et al* found that there was a preference for overweight applied to individuals of the opposite sex [72]. Comparing perceived body size to BMI indicated that obese men and women did not wish to gain weight and most obese or overweight women wanted to lose weight.

In a study which included service providers and refugees living in Geneva and not selected by BMI, Amstutz *et al* found that fruit and vegetables were considered healthy and that language and financial hardship were the main barriers to a healthy diet [94].

Alternative measures of adiposity were infrequently used with only thirteen studies recording waist circumference (WC) [59, 60, 63–66, 69, 70, 76, 79, 82, 84, 94].

### Risk of bias

S4 Appendix gives details of ROB for each study and Fig 4 summarises this across all studies. There was evidence of good practice in this challenging field, but only six studies were at low ROB overall and nine studies at moderate ROB. The challenge of achieving low ROB in the external domain was largely around choice of sampling frame and methods of participant selection. With internal ROB, the use of self-reported measures, definition of overweight and obesity and some unclear reporting were the main problems noted.

## Discussion

This review aimed to explore the prevalence and incidence of overweight and obesity, and the changes in adiposity over time in populations directly affected by humanitarian crises; the cascade of care in these populations and perceptions of patients with overweight and obesity. We included 56 reports derived from 45 studies. We found that prevalence estimates varied widely across the included studies and within subgroups based on study setting, internal ROB, exposure type and displacement status. Most studies report an increase in adiposity over time and compared to pre-exposure measures [47,48,63,78,79,87,90]. However, this relationship appears to be affected by displacement status. There were no reports detailing the cascade of care, but there is some evidence of limited physical exercise alongside a high calorie, low fruit and vegetable diet in refugee settings [50, 52, 57, 69]. We did not identify any studies in which the views of patients with obesity were sought qualitatively. However, a cross-sectional study did demonstrate cultural norms may differ in different settings [72].

### Burden of disease

Estimates of overweight range from 6.0% [56] to 53% [43]; for obesity from 0%[54] to 52.7% [51]; and for the two combined from 6.4% [56] to 82.8% [51]. These wide ranges persist in studies at low internal ROB. We did not identify any studies with no overweight and only one study with no obesity [54]. Whilst we were expecting to find overweight and obesity, we were surprised by the extent and ubiquitousness of the issue.

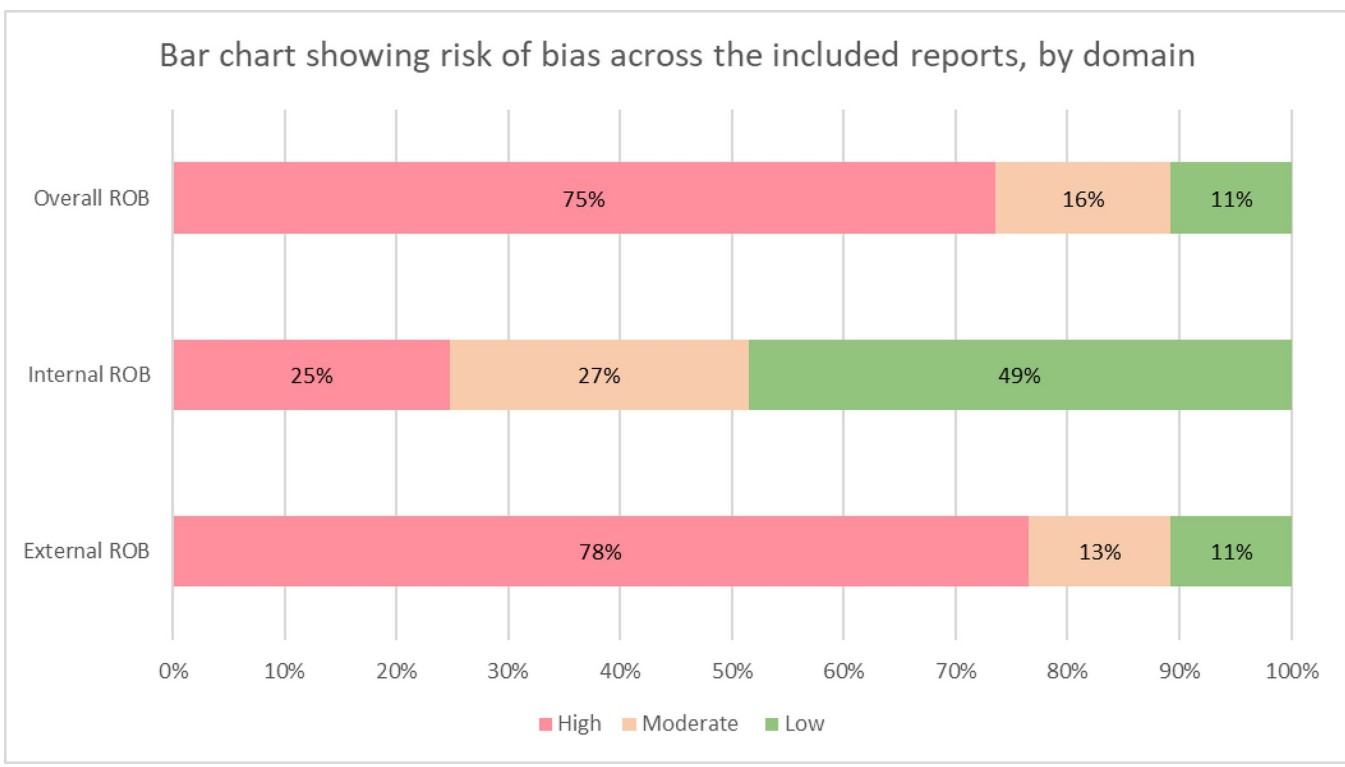

**Fig 4. Showing the risk of bias across the included reports, by domain.**

Generating a global prevalence estimate for obesity is complex. WHO estimates suggest that in 2016, 39% of adults were overweight and 13% were obese [97]. In the Global Burden of Disease (GBD) Study [39], estimates of overweight and obesity were higher in developed countries than developing countries in 2013. Our review would suggest that this pattern is not consistently seen in crisis-affected populations. Japanese and South Korean populations are the subject of nearly half of the HIC papers and these countries have some of the lowest levels of obesity and overweight for high income countries in the world [98]. For other HIC studies, populations came from LICs and were likely to be faced with poverty and other challenges in their new settings.

The GBD study points out that BMI tends to reach a peak at around 55 years in men and 60 years in women and that more women than men have a BMI greater than 25 kg/m$^2$ [39]. Several studies in this review formally tested the change in obesity and overweight estimates with increasing age and by sex. With regards to age, there was a consistent relationship between increasing age and increasing adiposity [45, 51, 52, 57, 59, 61, 69, 72, 93, 94]. With regards to sex, women were commonly found to have a higher rates of overweight and / or obesity than men in most reports [51, 52, 69, 71, 72, 93, 94]. However, it would appear that in some populations sexes are differentially affected by overweight and obesity [57, 70]. All except two [70, 71] of these reports had low internal ROB. These findings suggest that service providers can expect to find more overweight and obesity in older adults and females within a crisis affected population. The heterogeneity of our studies and the moderate to high external ROB means that it is difficult to generalise the extent of these differences.

Longitudinal changes in BMI are a function of age (as described above) and are part of the migration experience [99]. In migrant populations more generally, an initial health advantage

is superseded by increased risk of overweight and obesity compared to the native population. These changes are dependent on where the migrant comes from and how long they remain in the host country [100–102] All the studies reporting on these changes involved displaced populations making it difficult to comment on the differential effects of exposure to crises, acculturation and secular trends. Only five reports commenting on longitudinal changes were at low internal ROB [63, 66, 84, 94, 95]. Two reports from the NORNS study show that increasing duration in South Korea was associated with increase in weight, but that in some individuals weight loss is seen after the initial settling period [63, 66]. The Modesti report, which included only males, had a relatively short follow up period of 30 months which may explain why there was no significant change in adiposity seen [95]. And Ebner *et al* reported on individuals who had returned home after a relatively short displacement which may explain the change in trajectory of weight gain over time [84].

The Ohira et al papers have generated hazard ratios which show an increased risk of overweight and / or obesity with exposure to earthquakes [78, 79]. It is tempting to interpret this as evidence of a causal link between exposure to earthquakes and weight gain. However, the studies used observational data and a causal framework was not specified.

These findings would suggest that service providers, particularly in protracted situations, need to be prepared for increasing levels of overweight and obesity and the cardiometabolic complications that come with this. Whilst we do not propose that overweight and obesity are addressed in the immediate aftermath of a crisis, this pattern of weight gain points to an opportunity to take preventative action early in the time frame of a crisis.

As seen in other reviews of crisis affected populations [16, 24, 25], the geographical distribution of the studies and the people being examined is skewed. There is a long history of measuring adiposity as part of monitoring the impact of food aid. At the full text screening stage, we found that 63 reports did carry out anthropometric measurements, but that results were either presented in a way which did not allow categorisation or only those who were underweight or malnourished were reported (See S2 Appendix). This does mean that our range of prevalence estimates may have lower minimum bounds than we have identified. It also suggests that changing monitoring and reporting requirements would provide more information about the true prevalence of overweight and obesity and would clarify targets where more research would be most beneficial.

## Cascade of care

We could not identify evidence of information or interventions being directed specifically at those who were overweight or obese. It is likely that this reflects a genuine lack of interventions aimed at weight loss rather than NCD management more generally. However, there is evidence that, particularly those studies following WHO STEPS processes [103], were able to identify NCD risk factors. This provides a starting point for the discussion about targets for primary and secondary prevention. Namely, access to low calorie, high nutritional value food and promotion of active lifestyles [50, 52, 57, 69].

Looking at the cascade of care in NCD management more broadly, several recurring research and information gaps are noted. There is generally poor collection of standard data regarding disease states and recognised risk factors, there is a paucity of evidence to guide interventions, and there are infrastructure and supply problems even for those conditions in which treatments are available [16, 17, 31, 104]. Many of these factors are applicable to overweight and obesity. With the additional challenge that overweight and obesity are considered much later in the crisis response [15], by which time resources are arguably too stretched to extend to further activities.

## Patient perceptions

We did not identify qualitative studies in which we could differentiate the voices of those with overweight and obesity from other participants. However, cultural ideals and norms in relations to body size and shape were noted [72]. This is echoed particularly in work examining the understanding of African refugees who described the pursuit of thinness as perplexing [105]. Language and financial barriers to seeking care for overweight and obesity and also part of the refugee and migrant experience of seeking health care in general [106, 107].

Crisis affected populations are largely city dwellers [6] and as such multi-pronged and multi-level interventions are needed for both prevention and treatment [108]. However, it is acknowledged that population level weight loss interventions are challenging to implement and sustain even in well-resourced settings [109]. Causal pathways in obesity are complex [109]. Qualitative work is key to understanding the causal relationships between perceptions, understanding and behaviour. We cannot expect to successfully influence disease trajectories without this information.

## Use of waist circumference

We were surprised that only13 studies recorded WC [59, 60, 63–66, 69, 70, 76, 79, 82, 84, 94]. There is ongoing discussion about the most appropriate measure to determine increased adiposity [110]. Waist circumference provides important additional information in assessing the risk of death and disease due to increased adiposity [111]. However, WC is no longer explicitly mentioned WHO's Package of essential non-communicable disease interventions [112].

## Risk of bias

ROB poses a challenge in the interpretation of reported results. Identifying representative samples in crisis settings is a challenge, particularly with the chaos associated with displacement. The majority of studies weighed and measured their participants directly or used health records where these measurements were recorded. Several of the papers, however, used self reported heights and weights. These were coded as having a high internal ROB given the potentially inaccurate measurements. In Bhatta *et al*'s 2015 report, 7 people reported themselves to be overweight but the BMI data showed 70 people out of 120 to be overweight; the difference being a factor of 10 [59].

## Strengths and Limitations

One of the main strengths of this review is the number of reports included in the final analysis. These were identified in a systematic manner across databases and online repositories. The reports cover different crises and different regions of the world. Though this gives our analysis breadth, this heterogeneity means that we were unable to perform a meta-analysis and that the findings are not generalisable across all settings.

We used simple mathematics to derive missing prevalence estimates (marked in bold in Tables 2 and 3). In some cases, this involved calculations across multiple subgroups. This approach was taken as a pragmatic alternative to requesting access to individual patient data.

We only included publications from Jan 2011 onwards. On one hand this is a strength as it allows for a contemporaneous picture to emerge. On the other hand, it could be viewed as a weakness, since we will be missing patterns in change over time.

We did not identify reports discussing the cascade of care or the perception of patients with overweight or obesity. We believe that this genuinely reflects a paucity of data of this type. However, an alternative search strategy may have yielded different results. For example,

searching for the study type in the settings of interest and then screening for the disease could unearth different information.

## Conclusion

This study has shown that the prevalence of overweight and obesity vary in crisis affected populations but are rarely absent. Increases in adiposity over time, in older adults and in women are likely to be seen in most populations. Better quality descriptive information would help to identify precisely to who and when interventions should be offered in different settings. The lack of information about the cascade of care likely reflects limited efforts to address overweight and obesity in these settings. The lack of qualitative research hampers our understanding of which interventions would be most likely to succeed. WC measures should be included as part of standard care.

## Supporting information

**S1 Checklist. PRISMA 2020 main checklist.**
(DOCX)

**S1 Appendix. Full search strategies.**
(DOCX)

**S2 Appendix. Studies excluded at full text screening.**
(DOCX)

**S3 Appendix. Prevalence ranges in subgroups.**
(DOCX)

**S4 Appendix. Details of risk of bias assessment.**
(DOCX)

## Author Contributions

**Conceptualization:** Saran Shantikumar, Ammar Sabouni, Farah Kidy.

**Data curation:** Majel McGranahan, Farah Kidy.

**Methodology:** Thomas Shortland, Majel McGranahan, Daniel Stewart, Oyinlola Oyebode, William Proto, Ammar Sabouni, Farah Kidy.

**Project administration:** Farah Kidy.

**Supervision:** Oyinlola Oyebode, Saran Shantikumar, Farah Kidy.

**Visualization:** Gavin Rudge.

**Writing – original draft:** Thomas Shortland.

**Writing – review & editing:** Majel McGranahan, Daniel Stewart, Oyinlola Oyebode, Saran Shantikumar, William Proto, Bassit Malik, Roger Yau, Maddie Cobbin, Ammar Sabouni, Farah Kidy.

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
