## [Decision Letter · Decision Letter 0]

21 Jun 2022

PONE-D-22-11727A systematic review of the burden of overweight and obesity, access to services and patient views in humanitarian crisis settings.PLOS ONE

Dear Dr. Kidy,

Thank you for submitting your manuscript to PLOS ONE. After careful consideration, we feel that it has merit but does not fully meet PLOS ONE’s publication criteria as it currently stands. Therefore, we invite you to submit a revised version of the manuscript that addresses the points raised during the review process.

We look forward to receiving your revised manuscript.

Kind regards,

Che Matthew Harris

Academic Editor

PLOS ONE

Journal Requirements:

4. We note that Figure 2 in your submission contain [map/satellite] images which may be copyrighted. All PLOS content is published under the Creative Commons Attribution License (CC BY 4.0), which means that the manuscript, images, and Supporting Information files will be freely available online, and any third party is permitted to access, download, copy, distribute, and use these materials in any way, even commercially, with proper attribution. For these reasons, we cannot publish previously copyrighted maps or satellite images created using proprietary data, such as Google software (Google Maps, Street View, and Earth). For more information, see our copyright guidelines: http://journals.plos.org/plosone/s/licenses-and-copyright.

a. You may seek permission from the original copyright holder of Figure(s) [#] to publish the content specifically under the CC BY 4.0 license.  

Reviewers' comments:

Reviewer's Responses to Questions

**Comments to the Author**

1. Is the manuscript technically sound, and do the data support the conclusions?

Reviewer #1: Partly

2. Has the statistical analysis been performed appropriately and rigorously? 

Reviewer #1: N/A

3. Have the authors made all data underlying the findings in their manuscript fully available?

Reviewer #1: Yes

4. Is the manuscript presented in an intelligible fashion and written in standard English?

Reviewer #1: Yes

5. Review Comments to the Author

Reviewer #1: The aim of this review was to explore the prevalence of overweight and obesity in populations directly affected by humanitarian crises; the cascade of care in these populations and patients' knowledge and attitudes towards overweight and obesity.

The authors developed a systematic review considering the population and exposure (context) to perform the literature search. Although the systematic review is well designed, the authors establish many primary and secondary outcomes, which are not well discussed or exposed in the results, which weakens the article.

The main findings are that the prevalence of overweight and obesity varies considerably between studies, but an increase in adiposity was observed over time, in the elderly and in women. Most studies had a high risk of bias and few studies measured obesity beyond BMI (with waist circumference, for example). The other outcomes expected by the authors were poorly described.

TITLE

The article title does not make it clear whether access to services and patients' views are also related to obesity and overweight.

The term patient's views do not seem adequate for the proposed objective. I suggest changing to perceptions or knowledge. The context of the study in relation to the humanitarian crisis is unclear (are the results evaluated after, during, or at both times of a humanitarian crisis? Is it a before and after comparison?).

ABSTRACT

Aiming properly answer the objectives of the study, it would be important to add results and conclusions regarding the other two proposed objectives (i.e., the cascade of care in these populations and patient knowledge and attitudes to overweight and obesity) in the results of the abstract (lines 44-57).

INTRODUCTION

The last two sentences of the first paragraph of the introduction (lines 66-69) are repetitive and can be rewritten to improve understanding.

The second paragraph (lines 71-73) does not add relevant information for the development of the introduction and the line of reasoning on the topic proposed by the article. Thus, I suggest that it be rethought or replaced by a phrase that links obesity and reasons why it would increase/change in humanitarian crises.

Lines 99-101: as it appears as a primary outcome (line 151 of the Methods section) consider including incidence as an objective. Please, also consider the same regarding the change in adiposity over time (line 153 of the Methods section).

METHODS

Lines 104-108: was the review protocol registered in the International prospective register of systematic reviews (PROSPERO)? PROSPERO register is strongly endorsed prior to submitting manuscripts for publication, as recommended by the Cochrane guidelines for conducting systematic reviews. If there is a protocol number, please add it to the article. Protocol documents may also be uploaded as Supporting Information or linked from the Methods section of the article.

Lines 110-115: eligibility criteria require further clarification. While reasonable/rational, it is unclear (reading this section alone) whether the PECO was used for eligibility criteria. I recommend creating a subsection explaining the PECO of the study and then including a sentence in the eligibility criteria that makes it clear that the studies were also eligible by the PECO criteria.

Lines 113-114: what the authors meant by “Assessment of the returns showed that much of the earlier data were no longer applicable.” It is necessary to explain better the reason for which the authors reached this conclusion (Why does it not make sense? Why is it no longer applicable? What change has taken place?). If there is no plausible explanation, the restriction to review only publications from the last 10 years may not make sense and be considered a bias and should be reported in the study limitations.

Line 134: the exposure/context investigated in the study is unclear and needs to be improved. Do the authors intend to assess the aforementioned outcomes before, during, and after a humanitarian crisis? Just during a crisis? Or just after the crisis? Is it a before and after comparison? Even if the authors do not use it as inclusion/exclusion criteria, this information is important and seeks clarification.

Line 182: please add geographic localization in the data collection section as it appears in the methods of synthesis.

It is important to add, at some point in the text, the reasons why the authors decided to divide data into HIC and LMIC.

Lines 235-236: how do the authors divide countries into HIC and LMIC. Which parameter do they use to do this division? Please add a reference.

RESULTS

Figure 1: the flowchart needs clarification. Of 20,376 records screened, 19,885 were excluded. It should return 491 reports sought for retrieval instead of 488. In addition, consider reporting the number of records identified from each database rather than the total number across all databases.

Lines 242-244: it is not clear if this information (43 reports) should appear in the flowchart or if it is additional information since it does not match any number from the box “Reports excluded”.

Table 1: consider reviewing the use of acronyms (and putting them in the figure legend), and standardizing the use or not of full stops and the type of letter/font used in the table.

Tables 1, 2, and 3: for ease of reading, I suggest that the authors change the order of the articles described in Table 1, placing HICs articles first and not LMICs articles for each subgroup, according to the order in which they appear in Table 2 and Table 3. Another option is to place table 3 on table 2 and vice versa.

Table 2: how did you calculate some data which are all in bold (e.g. Kim et al, 2016 [61]; Ohira et al, 2016 [74]; and Takahashi et al, 2021 [79])?

Line 454: how much do the authors believe the search strategy influenced the result presented in the "other results" section? By combining obesity/overweight and humanitarian crisis, did you expect more results on the cascade of care and patient knowledge and attitudes about the disease? Do the authors believe that this low number of studies investigating these outcomes is due to a gap in the literature or to the search strategy?

It is not clear if these results are about people that already had pass by a humanitarian crisis or if they still is in humanitarian crisis. It should be interesting understanding if there is some difference between articles who investigated people that are still in the crisis or passed by a crisis.

DISCUSSION

Lines 489-490: although the authors found only results on obesity prevalence, it was not the only primary outcome of the study. Consider changing the first citation of the discussion to encompass all of the primary objectives initially proposed.

The discussion needs to be improved. It seems to me that the authors completely forgot about the last two primary outcomes of interest and focused only on the prevalence of overweight and obesity. Although the authors have not found results in this regard, they should discuss the gaps in the literature. The discussion lacks limitations of findings, as well as its strengths, perspectives, and needs related to the humanitarian crisis context. What is already known in the literature about the cascade of care in populations affected by humanitarian crises, even if it is related to another disease? What did you expect when investigating patient knowledge and attitudes towards overweight and obesity? Why do you assume it has not been reached?

CONCLUSION

Conclusions regarding primary outcomes of the review (number 3 and 4, the cascade of care and patient's knowledge and attitudes regarding the disease, respectively) are lacking.

The conclusion could be more concise, pointing out the main findings, and implications to clinical practice (to health professionals, patients care, and health services) and research (due to low and limited evidence).

6. PLOS authors have the option to publish the peer review history of their article (what does this mean?). If published, this will include your full peer review and any attached files.

Reviewer #1: No

---

## [Author Response · Author response to Decision Letter 0]

11 Jan 2023

Dear Editorial team and Reviewer, 

Re: PONE-D-22-11727. A systematic review of the burden of, access to services for and perceptions of patients with overweight and obesity in humanitarian crisis settings.

Thank you very much for taking the time to review our manuscript and for providing useful feedback. Please find responses in the tables below.

Table One: Journal requirements

Please ensure that your manuscript meets PLOS ONE's style requirements, including those for file naming. The paper has been reformatted to meet the style requirements. 

In your Data Availability statement, you have not specified where the minimal data set underlying the results described in your manuscript can be found. PLOS defines a study's minimal data set as the underlying data used to reach the conclusions drawn in the manuscript and any additional data required to replicate the reported study findings in their entirety. All PLOS journals require that the minimal data set be made fully available. 

Apologies, this was a misunderstanding on our part. All data used in this study were derived from published papers and so are already in the public domain. 

We note that you have indicated that data from this study are available upon request. PLOS only allows data to be available upon request if there are legal or ethical restrictions on sharing data publicly. All data underlying this study are in the public domain. Our data extraction sheets will be available on request. 

We note that Figure 2 in your submission contain [map/satellite] images which may be copyrighted. All PLOS content is published under the Creative Commons Attribution License (CC BY 4.0), which means that the manuscript, images, and Supporting Information files will be freely available online, and any third party is permitted to access, download, copy, distribute, and use these materials in any way, even commercially, with proper attribution. For these reasons, we cannot publish previously copyrighted maps or satellite images created using proprietary data, such as Google software (Google Maps, Street View, and Earth). For more information, see our copyright guidelines: http://journals.plos.org/plosone/s/licenses-and-copyright.

a. You may seek permission from the original copyright holder of Figure(s) [#] to publish the content specifically under the CC BY 4.0 license. 

The maps presented in Figure 2 were produced in R using ggplot with no additional map data imported. The authors of the software list R studio as the copyright holder (https://ggplot2.tidyverse.org/authors.html). 

The Creative Commons Attribution by 4.0 license is viewable here and has been added to the submission as suggested. https://www.r-project.org/Licenses/CC-BY-SA-4.0

Table two: Reviewer Comments

TITLE:

The article title does not make it clear whether access to services and patients' views are also related to obesity and overweight.

The term patient's views do not seem adequate for the proposed objective. I suggest changing to perceptions or knowledge. 

Thank you for your suggestion. “Patient views” have been replaced with “Patient perceptions”. The title has been reordered to read “A systematic review of the burden of, access to services for and perceptions of patients with overweight and obesity in humanitarian crisis settings.”

This description of patient perceptions has been repeated through the paper. 

The context of the study in relation to the humanitarian crisis is unclear (are the results evaluated after, during, or at both times of a humanitarian crisis? Is it a before and after comparison?). Thank you for raising this point. The temporal relationship between data collection and exposure varied across the studies. We have addressed this point in more detail below. In order to prevent the title from becoming too complex, we have retained the broad description of "humanitarian crises”

ABSTRACT:

Aiming properly answer the objectives of the study, it would be important to add results and conclusions regarding the other two proposed objectives (i.e., the cascade of care in these populations and patient knowledge and attitudes to overweight and obesity) in the results of the abstract (lines 44-57) 

The results section has been amended to read, “There were no studies reporting cascade of care. No qualitative studies were identified.”

The conclusion has been enhanced to read: “Gaps remain in understanding the existing cascade of care. Cultural norms around diet and ideal body size vary.” 

INTRODUCTION

The last two sentences of the first paragraph of the introduction (lines 66-69) are repetitive and can be rewritten to improve understanding. 

The last two sentences have been edited to avoid repetition. 

The second paragraph (lines 71-73) does not add relevant information for the development of the introduction and the line of reasoning on the topic proposed by the article. Thus, I suggest that it be rethought or replaced by a phrase that links obesity and reasons why it would increase/change in humanitarian crises. 

Thank you for raising this issue. This paragraph had been included to highlight that obesity and overweight have potential negative consequences for both communicable and non-communicable disease and therefore the study of adiposity remains important even in the time of Covid. A sentence has been added to the start of the paragraph to clarify this. 

After describing where most of the displaced people in the world have come from, we have added a sentence to point out epidemiological transition is already underway in many of these countries and that levels of adiposity may increase or decrease depending on the crisis. 

Lines 99-101: as it appears as a primary outcome (line 151 of the Methods section) consider including incidence as an objective. Please, also consider the same regarding the change in adiposity over time (line 153 of the Methods section) 

These changes have been made. 

METHODS

Lines 104-108: was the review protocol registered in the International prospective register of systematic reviews (PROSPERO)? PROSPERO register is strongly endorsed prior to submitting manuscripts for publication, as recommended by the Cochrane guidelines for conducting systematic reviews. If there is a protocol number, please add it to the article. Protocol documents may also be uploaded as Supporting Information or linked from the Methods section of the article. Thank you for raising this point. Unfortunately, we were unable to register this review on Prospero. 

At the time the review was being conducted, there was a change in policy which only allowed registration of protocols which would include a metanalysis. Our scoping study and previous work had demonstrated that the heterogeneity of studies in this field would make a metanalysis challenging. 

Lines 110-115: eligibility criteria require further clarification. While reasonable/rational, it is unclear (reading this section alone) whether the PECO was used for eligibility criteria. I recommend creating a subsection explaining the PECO of the study and then including a sentence in the eligibility criteria that makes it clear that the studies were also eligible by the PECO criteria. 

Thank you for this suggestion. We have added a subsection for the PECO criteria and have clarified that these are also part of the eligibility criteria. 

Lines 113-114: what the authors meant by “Assessment of the returns showed that much of the earlier data were no longer applicable.” It is necessary to explain better the reason for which the authors reached this conclusion (Why does it not make sense? Why is it no longer applicable? What change has taken place?). If there is no plausible explanation, the restriction to review only publications from the last 10 years may not make sense and be considered a bias and should be reported in the study limitations. 

Thank you for pointing this out. We have tried to clarify our position with this explanation: “Reviewing the returns showed that the data being presented in the earlier papers were out of date given the context of changing levels of obesity globally. Since we were interested in providing a description which could be used by service providers in the current time, we restricted this review to papers published from January 1st, 2011, onwards.”

To be transparent we have added a sentence to the strengths and limitations section to highlight both the positive and negative aspects of this decision. 

Line 134: the exposure/context investigated in the study is unclear and needs to be improved. Do the authors intend to assess the aforementioned outcomes before, during, and after a humanitarian crisis? Just during a crisis? Or just after the crisis? Is it a before and after comparison? Even if the authors do not use it as inclusion/exclusion criteria, this information is important and seeks clarification. 

We agree that this is an important point. We have added the following sentences to the PECO description: “Exposures needed to be ongoing or previous to the time of data collection. We did not impose other temporal restrictions on the exposure-outcome relationship.”

Line 182: please add geographic localization in the data collection section as it appears in the methods of synthesis. 

This has been added. 

It is important to add, at some point in the text, the reasons why the authors decided to divide data into HIC and LMIC.

Lines 235-236: how do the authors divide countries into HIC and LMIC. Which parameter do they use to do this division? Please add a reference. 

We have added the following plus a reference to address this issue: “Categorisation as HIC or LMIC was selected to allow comparison to other publications in this field and to allow a rough assessment of resources available at a country level. The World Bank income-based classification system was used.”

RESULTS

Figure 1: the flowchart needs clarification. Of 20,376 records screened, 19,885 were excluded. It should return 491 reports sought for retrieval instead of 488. 

Thank you for spotting this. We have checked through our data flow and the error is in the number excluded. This has been changed to 19,888.

In addition, consider reporting the number of records identified from each database rather than the total number across all databases. 

This has now been added. The 2019 searches were part of our scoping exercise, hence the numbers in the new flow chart also includes those returns which were excluded once the new cut-off date was applied. 

Lines 242-244: it is not clear if this information (43 reports) should appear in the flowchart or if it is additional information since it does not match any number from the box “Reports excluded”. 

Thank you for pointing this out. We have edited to bring in line with the categorisations used in the flow diagram. 

Table 1: consider reviewing the use of acronyms (and putting them in the figure legend), and standardizing the use or not of full stops and the type of letter/font used in the table. These changes have been made. 

Tables 1, 2, and 3: for ease of reading, I suggest that the authors change the order of the articles described in Table 1, placing HICs articles first and not LMICs articles for each subgroup, according to the order in which they appear in Table 2 and Table 3. Another option is to place table 3 on table 2 and vice versa. 

Thank you for this suggestion. We have made the LMIC table into Table 2 and the HIC Table into Table 3. 

Table 2: how did you calculate some data which are all in bold (e.g. Kim et al, 2016 [61]; Ohira et al, 2016 [74]; and Takahashi et al, 2021 [79])? 

For some papers, results were reported for subgroups only. Where we felt subgroups were mutually exclusive, we summed the data presented to give whole sample information. Where results were given as rates, the rates were applied to population of the appropriate subgroup to give the numbers in each subgroup. Then the summing procedure was repeated. 

An explanation has been added to the methods section. 

Line 454: how much do the authors believe the search strategy influenced the result presented in the "other results" section? By combining obesity/overweight and humanitarian crisis, did you expect more results on the cascade of care and patient knowledge and attitudes about the disease? Do the authors believe that this low number of studies investigating these outcomes is due to a gap in the literature or to the search strategy? 

Thank you for raising this issue. We were keen to capture a broad range of study types exploring different aspects of overweight and obesity. Our search strategy did include terms for both non-communicable diseases and chronic diseases as well as obesity and overweight. This approach was taken so that we could screen returns to see if overweight and obesity were being considered alongside other NCDs. However, to be included, we needed to be able to pick out information about these states from the information pertaining to other NCDs. We found however, that most broad NCD papers did not include data about obesity. 

It is however possible that a different approach, for example searching by study type and setting and then manually screening for overweight and obesity could have yielded different results. We have included this point in our discussion section. 

It is not clear if these results are about people that already had pass by a humanitarian crisis or if they still is in humanitarian crisis. It should be interesting understanding if there is some difference between articles who investigated people that are still in the crisis or passed by a crisis. 

Thank you for raising this important point. All participants had been exposed to a crisis. For the studies exploring those exposed to natural disasters, the data collection point we have reported was carried out after the event. For the remainder, the participants were living in long term refugee situations or going through the asylum seeking / refugee resettlement processes. 

However, it is difficult to be clear on when the exposure has ended. For example, we are uncertain when, if ever, a long-standing refugee situation stops being a crisis from the perspective of the participant. Or when exposure to conflict or a natural disaster stops having an impact on health. 

We have added these points to the results: Data reported in this paper were all collected after the exposure had begun. For those in long-standing refugee situations or going through asylum seeking or refugee resettlement processes, the exposure was considered to be ongoing. For those exposed to natural disasters, data collection took place between 4 months (Herrera-Fontana et al. 2020) and 4 years after the disaster (Satoh et al. 2021; Takahashi et al. 2020). 

You had also raised the question about studies which compared changes before and after the disaster. We had not included this factor as one of our objectives. However, we agree that it adds an important nuance. We have edited the results section on “Changes over time and with displacement” to draw out these findings. The following paragraphs have been added: “Four reports formally compared changes in adiposity before and after exposure to the Great East Japan Earthquake (Hikichi et al. 2019; Ebner et al. 2016; Ohira et al. 2017; Ohira et al. 2016). Hikichi et al report that approximately 2.5 years after the disaster, the prevalence of obesity had increased amongst those displaced (25.0% to 35.1%) but decreased amongst those not displaced (26.9% to 26.6%) compared to 7 months before the disaster (Hikichi et al. 2019). Ebner et al report that the OR of obesity was higher in the year after the disaster, but that this risk was no longer significant in the second year after the disaster (OR 1.31 (95% CI 1.06 to 1.38) and 1.07 (95% CI 0.93 to 1.24) respectively).(Ebner et al. 2016)

Ohira et al report that BMI and obesity increased in earthquake affected populations. This increase was greater in those evacuated compared to those not evacuated and greater in males compared to females. The multivariable adjusted hazard ratio for overweight after the disaster was 1.61 (95% CI 1.47- 1.77). (Ohira et al. 2016; Ohira et al. 2017). 

Only one non-earthquake study compared BMI before and after exposure. No change was found. (Kory et al. 2013)

DISCUSSION

Lines 489-490: although the authors found only results on obesity prevalence, it was not the only primary outcome of the study. Consider changing the first citation of the discussion to encompass all of the primary objectives initially proposed. 

This line has been replaced to reflect the objectives as described in the introduction: “This review aimed to explore the prevalence and incidence of overweight and obesity, and the changes in adiposity over time in populations directly affected by humanitarian crises; the cascade of care in these populations and perceptions of patients with overweight and obesity.” The first paragraph has also been edited to include: “Most studies report an increase in adiposity over time and compared to pre-exposure measures [47,48,63,78,79,87,90]. However, this relationship appears to be affected by displacement status. There were no reports detailing the cascade of care, but there is some evidence of limited physical exercise alongside a high calorie, low fruit and vegetable diet in refugee settings. [50,52,57,69] We did not identify any studies in which the views of patients with obesity were sought qualitatively. However a cross-sectional study did demonstrate cultural norms may differ in different settings. [72]” 

The discussion needs to be improved. It seems to me that the authors completely forgot about the last two primary outcomes of interest and focused only on the prevalence of overweight and obesity. Although the authors have not found results in this regard, they should discuss the gaps in the literature. The discussion lacks limitations of findings, as well as its strengths, perspectives, and needs related to the humanitarian crisis context. What is already known in the literature about the cascade of care in populations affected by humanitarian crises, even if it is related to another disease? What did you expect when investigating patient knowledge and attitudes towards overweight and obesity? Why do you assume it has not been reached? We are sorry to hear that our discussion section was found to be lacking. We have added subheadings to ensure that all objectives are covered. 

With regards to prevalence estimates, changes over time and impacts of displacement, we feel that these points have been well covered. We have added the following paragraph to caution against over-interpretation of the data we have added comparing pre-and post-disaster measures: 

“The Ohira et al papers have generated hazard ratios which show an increased risk of overweight and / or obesity with exposure to earthquakes. (Ohira et al. 2016; Ohira et al. 2017) It is tempting to interpret this as evidence of a causal link between exposure to earthquakes and weight gain. However, the studies used observational data and a causal framework was not specified.”

With regards to cascade of care, we have added a comment that we believe the lack of data reflects a lack of activity in this sphere. We have also included a paragraph describing the situation with other NCDs. “Looking at the cascade of care in NCD management more broadly, several recurring research and information gaps are noted. There is generally poor collection of standard data regarding disease states and recognised risk factors, there is a paucity of evidence to guide interventions, and there are infrastructure and supply problems even for those conditions in which treatments are available. [16,17,31,105] Many of these factors are applicable to overweight and obesity. With the additional challenge that overweight and obesity are considered much later in the crisis response, [15] by which time resources are arguably too stretched to extend to further activities.”

With regards to patient perceptions, we have reorganised the text and added further details: “Crisis affected populations are largely city dwellers [6] and as such multi-pronged and multi-level interventions are needed for both prevention and treatment.[103] Causal pathways in obesity are complex [106] and are likely to be further complicated by exposure to crises. Qualitative work is key to understanding the causal relationships between perceptions, understanding and behaviour. We cannot expect to successfully influence disease trajectories without this information.”

Conclusions regarding primary outcomes of the review (number 3 and 4, the cascade of care and patient's knowledge and attitudes regarding the disease, respectively) are lacking. Information about these outcomes has been added 

The conclusion could be more concise, pointing out the main findings, and implications to clinical practice (to health professionals, patients care, and health services) and research (due to low and limited evidence). 

The conclusion has been rewritten to state: “This study has shown that the prevalence of overweight and obesity vary in crisis affected populations but are rarely absent. Increases in adiposity over time, in older adults and in women are likely to be seen in most populations. Better quality descriptive information would help to identify precisely who and when interventions should be offered in different settings. The lack of information about the cascade of care likely reflects limited efforts to address overweight and obesity in these settings. The lack of qualitative research hampers our understanding of what interventions would be most likely to succeed. WC measures should be included as part of standard care.

Thank once again for taking the time to give us feedback on this piece of work. The suggestions have resulted in increased clarity and readability. We hope that you find our responses satisfactory. 

Kind regards

Farah Kidy

---

## [Decision Letter · Decision Letter 1]

10 Feb 2023

PONE-D-22-11727R1A systematic review of the burden of, access to services for and perceptions of patients with overweight and obesity in humanitarian crisis settingsPLOS ONE

Dear Dr. Kidy,

Thank you for submitting your manuscript to PLOS ONE. After careful consideration, we feel that it has merit but does not fully meet PLOS ONE’s publication criteria as it currently stands. Therefore, we invite you to submit a revised version of the manuscript that addresses the points raised during the review process.

ACADEMIC EDITOR: I noticed that there may be minor errors in the reference order.  It looks as if references 104 and 105 (page 38) now comes before reference 103 (page 39) in the updated version. Similarly references 107-109 come before 106 in the revised version. Please review and correct references where needed. Thank you.==============================

We look forward to receiving your revised manuscript.

Kind regards,

Che Matthew Harris

Academic Editor

PLOS ONE

Journal Requirements:

Reviewers' comments:

Reviewer's Responses to Questions

**Comments to the Author**

1. If the authors have adequately addressed your comments raised in a previous round of review and you feel that this manuscript is now acceptable for publication, you may indicate that here to bypass the “Comments to the Author” section, enter your conflict of interest statement in the “Confidential to Editor” section, and submit your "Accept" recommendation.

Reviewer #1: All comments have been addressed

2. Is the manuscript technically sound, and do the data support the conclusions?

Reviewer #1: Yes

3. Has the statistical analysis been performed appropriately and rigorously? 

Reviewer #1: N/A

4. Have the authors made all data underlying the findings in their manuscript fully available?

Reviewer #1: Yes

5. Is the manuscript presented in an intelligible fashion and written in standard English?

PLOS ONE does not copyedit accepted manuscripts, so the language in submitted articles must be clear, correct, and unambiguous. Any typographical or grammatical errors should be corrected at revision, so please note any specific errors here. T

Reviewer #1: Yes

6. Review Comments to the Author

Reviewer #1: (No Response)

7. PLOS authors have the option to publish the peer review history of their article (what does this mean?). If published, this will include your full peer review and any attached files.

Reviewer #1: No

---

## [Author Response · Author response to Decision Letter 1]

23 Feb 2023

Dear Che and team, 

Re: PONE-D-22-11727R1

A systematic review of the burden of, access to services for and perceptions of patients with overweight and obesity in humanitarian crisis settings

Thank you taking the time to review the revision of our paper. We are pleased that the reviewer feels that all previous points have been addressed. 

Further response below:

I noticed that there may be minor errors in the reference order. It looks as if references 104 and 105 (page 38) now comes before reference 103 (page 39) in the updated version. Similarly references 107-109 come before 106 in the revised version. Please review and correct references where needed. Thank you.

Thank you for raising this point. We have checked and updated all the references, and these are now in the correct order. 

Thank you once again

Farah Kidy

---

## [Editor Report · Decision Letter 2]

24 Feb 2023

A systematic review of the burden of, access to services for and perceptions of patients with overweight and obesity in humanitarian crisis settings

PONE-D-22-11727R2

Dear Dr. Kidy,

We’re pleased to inform you that your manuscript has been judged scientifically suitable for publication and will be formally accepted for publication once it meets all outstanding technical requirements.

Kind regards,

Che Matthew Harris

Academic Editor

PLOS ONE
---

## [Editor Report · Acceptance letter]

13 Apr 2023

PONE-D-22-11727R2 

A systematic review of the burden of, access to services for and perceptions of patients with overweight and obesity, in humanitarian crisis settings. 

Dear Dr. Kidy:

I'm pleased to inform you that your manuscript has been deemed suitable for publication in PLOS ONE. Congratulations! Your manuscript is now with our production department. 

Kind regards, 

on behalf of

Dr. Che Matthew Harris 

Academic Editor

PLOS ONE